# The β-1,4 GalT-V Interactome—Potential Therapeutic Targets and a Network of Pathways Driving Cancer and Cardiovascular and Inflammatory Diseases

**DOI:** 10.3390/ijms26168088

**Published:** 2025-08-21

**Authors:** Subroto Chatterjee, Dhruv Kapila, Priya Dubey, Swathi Pasunooti, Sruthi Tatavarthi, Claire Park, Caitlyn Ramdat

**Affiliations:** The Helen B Taussig Heart Center, Cardiovascular Innovation Laboratory, Division of Cardiology, Department of Pediatrics, Johns Hopkins University School of Medicine, Baltimore, MD 21287, USA; dkapila1@jhu.edu (D.K.); pdubey5@jh.edu (P.D.); spasuno1@jh.edu (S.P.); statava1@jh.edu (S.T.); cpark62@jh.edu (C.P.);

**Keywords:** GalT-V, lactosylceramide, specific protein-1, Notch-1, cancer

## Abstract

UDP-Gal-β-1,4 galactosyltransferase-V (GalT-V) is a member of a large family of galactosyltransferases whose function is to transfer galactose from the nucleotide sugar UDP-galactose to a glycosphingolipid glucosylceramide, to generate lactosylceramide (LacCer). It also causes the N and O glycosylation of proteins in the Trans Golgi area. LacCer is a bioactive lipid second messenger that activates an “oxidative stress pathway”, leading to critical phenotypes, e.g., cell proliferation, migration angiogenesis, autophagy, and apoptosis. It also activates an “inflammatory pathway” that contributes to the progression of disease pathology. β-1,4-GalT-V gene expression is regulated by the binding of the transcription factor Sp-1, one of the most O-GlcNAcylated nuclear factors. This review elaborates the role of the Sp-1/GalT-V axis in disease phenotypes and therapeutic approaches targeting not only Sp-1 but also Notch-1, Wnt-1 frizzled, hedgehog, and β-catenin. Recent evidence suggests that β-1,4GalT-V may glycosylate Notch-1 and, thus, regulate a VEGF-independent angiogenic pathway, promoting glioma-like stem cell differentiation into endothelial cells, thus contributing to angiogenesis. These findings have significant implications for cancer and cardiovascular disease, as tumor vascularization often resumes aggressively following anti-VEGF therapy. Moreover, LacCer can induce angiogenesis independent of VEGF and its level are reported to be high in tumor tissues. Thus, targeting both VEGF-dependent and VEGF-independent pathways may offer novel therapeutic strategies. This review also presents an up-to-date therapeutic approach targeting the β-1,4-GalT-V interactome. In summary, the β-1,4-GalT-V interactome orchestrates a broad network of signaling pathways essential for maintaining cellular homeostasis. Conversely, its dysregulation can promote unchecked proliferation, angiogenesis, and inflammation, contributing to the initiation and progression of multiple diseases. Environmental factors and smoking can influence β-1,4-GalT-V expression and its interactome, whereas elevated β-1,4-GalT-V expression may serve as a diagnostic biomarker of colorectal cancer, inflammation—exacerbated by factors that may worsen pre-existing cancer malignancies, such as smoking and a Western diet—and atherosclerosis, amplifying disease progression. Increased β-1,4-GalT-V expression is frequently associated with tumor aggressiveness and chronic inflammation, underscoring its potential as both a biomarker and therapeutic target in colorectal and other β-1,4-GalT-V-driven cancers, as well as in cardiovascular and inflammatory diseases.

## 1. Introduction

UDP-Gal-β-1,4-galactosyltransferase-V belongs to a large family of galactosyl transferases. Although both β-1,4-GalT-V and β-1,4GalT-VI produce LacCer, this review focuses on β-1,4-GalT-V due to its emerging physiological relevance in health and disease. Historically, β-1,4-GalT-V has been associated with the biosynthesis of N-glycans, O-glycans, and a glycosphingolipid, LacCer. The presence of GlcNAc 1–6 branched mannose residues in N-glycans is a hallmark of tumor cells [1,2], lending credence to β-1,4-GalT-V’s role in cancer pathophysiology.

Interest in β-1,4-GalT-V intensified following studies showing increased expression of this gene and enzyme in various cancers. Functional studies have revealed that both β-1,4-GalT-V knockout and overexpression significantly altered tumor cell proliferation [3]. Recently, β-1,4-GalT-V has been implicated in stem cell differentiation and the glycosylation of Notch-1, a cancer- and angiogenesis-associated transmembrane protein (Figure 1). Accordingly, this review highlights the β-1,4-GalT-V interactome—i.e., the network of genes, proteins, and lipids with which β-1,4-GalT-V interacts—as they also serve as potential therapeutic targets in various diseases.

A breakthrough in understanding β-1,4-GalT-V regulation came with the discovery that several physiologically relevant stimuli—such as VEGF, PDGF, LDL/oxidized LDL, sheer stress, Western diet, and cigarette smoke—can activate β-1,4-GalT-V to generate LacCer. For example, oxidized LDL binds its receptor and triggers a kinase cascade that phosphorylates serine, threonine, and tryptophan and activates β-1,4-GalT-V in human arterial smooth muscle and kidney proximal tubular cells. Post-transcriptional phosphorylation of β-1,4-GalT-V enhances its enzymatic activity, promoting LacCer synthesis. LacCer acts as a bioactive lipid second messenger that regulates key cellular phenotypes, such as proliferation, migration, adhesion, angiogenesis, and apoptosis (Figure 1) [6]. Conversely, inhibition of β-1,4-GalT-V using agents, like D-PDMP, attenuates its downstream signaling effects, highlighting its therapeutic potential [7].

This review also aims to underscore that β-1,4-GalT-V exerts broad regulatory influence by glycosylating critical proteins—such as Notch-1—thereby affecting pathways tied to cell fate, differentiation, proliferation, and apoptosis. Given that Notch can function as an oncogene [8], its glycosylation by β-1,4-GalT-V is an area of pivotal interest.

Additionally, obesity—a rising epidemic linked to sedentary lifestyles and Western diet—has emerged as a key risk factor in colorectal cancer (CRC), as adipose tissue is metabolically active, secreting proinflammatory cytokines, disrupting insulin signaling, and possibly influencing epigenetic changes that predispose colon cells to malignancy.

Of emerging concern is the increasing incidence of CRC in young adults under 50, a group traditionally considered low risk. We speculate that the contributing factors may include high-fat diets, microbiome dysbiosis, and low dietary fiber. β-1,4-GalT-V has gained attention as a molecular contributor, with altered expression correlating with tumor aggressiveness and inflammation, making it a promising diagnostic or therapeutic target in early onset CRC as well as several other cancers wherein the β-1,4-GalT-V gene/protein are overexpressed. For example, we have shown overexpression of the gene, protein, and LacCer level in more tissues in clinically well-defined patients with CRC at the Johns Hopkins Hospital. More impressively, two pathologists using a monoclonal β-1,4-GalT-V antibody could not only independently diagnose CRC but also classified the patients into phases 1–4 of this disease [9]. Additionally, we have previously shown that a Western diet raises β-1,4-GalT-V protein expression, enzyme activity, and LacCer levels in ApoE-/- mice and rabbits. This is accompanied by arterial stenosis, vascular stiffness, cardiac hypertrophy, and cardiac dysfunction. And this was mitigated by blocking the activity of β-1,4-GalT-V [10,11].

These insights reinforce the notion that our lifestyle choices are not biologically neutral; they engage our genome and metabolism in complex, often enduring ways, influencing disease risk or resilience.

## 2. β-1,4-GalT-V and Glucosylceramide: Formation of LacCer

β-1,4-Galactosyltransferase V (β-1,4 GalT-V) catalyzes the galactosylation of glucosylceramide (GlcCer) to generate lactosylceramide (LacCer), a crucial intermediate in glycosphingolipid biosynthesis (Figure 1). This reaction is highly substrate-specific, as β-1,4-GalT-V shows limited activity with alternate ceramide homologs. LacCer acts as a potent bioactive lipid that activates the following:The oxidative stress pathway via superoxide (O_2_^−^) production.The inflammatory pathway by activating cytosolic phospholipase A2 (cPLA2), which releases arachidonic acid from phosphatidylcholine—an upstream precursor of proinflammatory prostaglandins.

In human arterial endothelial cells, VEGF activates β-1,4-GalT-V via its receptor, triggering LacCer production and promoting angiogenesis and tumorigenic signaling [9]. β-1,4-GalT-V dysregulation has, thus, been linked to conditions including atherosclerosis and colorectal cancer [9].

Multiple physiological stimuli—oxidized-LDL, VEGF, PDGF, EGF, shear stress, cigarette smoke, and proinflammatory cytokines, like TNF-α—converge to activate β-1,4-GalT-V and elevate LacCer levels. On the other hand, as a “bonafide” second messenger, LacCer independently promotes cell proliferation, migration, adhesion, angiogenesis, apoptosis, and phagocytosis. Aberrant β-1,4-GalT-V or LacCer signaling contributes to diseases such as cardiovascular disorders, diabetes, systemic lupus erythematosus, ischemia–reperfusion injury, and cancer. Inflammatory disorders such as COPD, inflammatory bowel disease, neuroinflammation, and certain skin conditions are similarly linked to LacCer signaling. For detailed analysis, refer to our prior review on LacCer-mediated inflammation [9].

## 3. NOX Activation by LacCer and Superoxide Production

Studies in cultured human arterial smooth muscle cells show that LacCer treatment dose- and time-dependently increases superoxide (O_2_^−^) production, by activating NADPH oxidase (NOX) by increasing the protein expression of p47phox and p92phox [12]. Thus p47 phox refers to NOX2 (phagocytic NADPH oxidase) and p92 phox refers to protein p22phox, essential for the activation of several NOX complexes, e.g., NOX1,3,4. Superoxide generation activates kinase pathways, including RAS, MAPK, AKT-1, and mTOR-c1, driving cell proliferation (Figure 1). In Chinese hamster ovary cells exclusively expressing β-1,4-GalT-V gene/protein, LacCer also activates PI3K/AKT signaling to induce cell proliferation [13].

LacCer activates platelet-endothelial cell adhesion molecule (PECAM-1) gene and protein expression in human endothelial cells, promoting angiogenesis [13]. In human neutrophils, LacCer also generates O_2_^−^ and activates cytosolic phospholipase-C (cPLA-2), which cleaves arachidonic acid from phosphatidylcholine. Arachidonic acid, a precursor to various prostaglandins, then activates inflammatory pathways. Additionally, in neutrophils, LacCer increases expression of the cell adhesion molecule CD11b at the cell surface, which serves as a receptor for PECAM-1/ICAM-1 on endothelial cells. This endothelial–neutrophil/monocyte adhesion is followed by the internalization/migration of these blood cells into the endothelium, contributing to exogenous LacCer and inflammation. A recent lipidomic study of monocytes from patients with acute myocardial infarction showed that LacCer may serve in monocyte migration [14].

LacCer enhances ROS production, supporting the innate immune response against infections and inflammation [15]. Elevated LacCer levels have been observed in various conditions. Superoxide generation, a key part of the oxidative stress response, stems from LacCer-induced signaling pathways, particularly in macrophages and neutrophils [9,15]. LacCer-rich lipid rafts facilitate essential signaling functions that drive superoxide production during immune responses. For example, LacCer mediates phagocytosis and neutrophil migration; in conditions such as pulmonary emphysema, this can promote aberrant autophagy and inflammatory responses [16]. In summary, while LacCer-induced low level ROS production supports the innate immune response and autophagy, excessive ROS can contribute to inflammation and cardiac hypertrophy [17,18].

## 4. β-1,4-GalT-V Inhibitors

D-threo-1-phenyl-2-decanoylamino-3-morpholino-1-propanol (D-PDMP) is a widely used small-molecule inhibitor of glycosphingolipid (GSL) biosynthesis. It primarily targets glucosylceramide synthase (GCS), which catalyzes the transfer of glucose from UDP-glucose to ceramide—the first glycosylation step in the GSL biosynthetic pathway.

Inhibiting GCS disrupts downstream GSL synthesis, affecting membrane dynamics, signaling, and cell interactions. D-PDMP is used in basic and translational studies of lipid metabolism, lysosomal storage diseases, and cancer. However, we have shown that D-PDMP can also inhibit β-1,4-GalT-V activity in pure enzyme preparations, cell-based assays, and multiple mouse and rabbit models of cardiovascular diseases and type II diabetes [10]. Additionally, in cell-based assays, D-PDMP-mediated inhibition of β-1,4- was bypassed by feeding LacCer more efficiently than GlcCer and downstream phenotypes, e.g., angiogenesis and cell migration.

In endothelial cells, β-1,4-GalT-V knockdown using siRNA reduces β-1,4-GalT-V expression and attenuates VEGF-induced angiogenesis [19]. Interestingly, LacCer supplementation bypasses this inhibition, suggesting a VEGF-independent β-1,4-GalT-V/LacCer pathway (further discussed in the β-1,4-GalT-V-Notch-1 section).

## 5. β-1,4-GalT-V-Protein Interactome

This section summarizes recent evidence that β-1,4-GalT-V interacts directly or indirectly with physiologically relevant proteins that regulate diverse cellular pathways and impact human health and disease.

### 5.1. β-1,4-GalT-V Interaction with Sp-1

Multifaceted Role of Sp1 in Tumor Progression: From Cell Cycle Regulation to pH-tome Modulation.

#### 5.1.1. Sp1 Regulates β-1,4-GalT-V Gene and Protein Expression

Sato and Furukawa identified four Sp1 binding sites within the promoter region of the β-1,4-GalT-V gene, reinforcing the role of Sp1 as a key transcriptional regulator. Mutation of the Sp1 binding site at nucleotide positions –81/–69 significantly reduced the promoter activity. Transfection of A549 cells with Sp1 cDNA increased promoter activity, while treatment with mithramycin A—a drug that blocks SP1–DNA-binding—suppressed this activation. These findings indicate that Sp1 directly regulates β-1,4-GalT-V transcription in cancer cells. The presence of multiple Sp1 binding motifs further amplifies its capacity to upregulate β-1,4-GalT-V expression, identifying β-1,4-GalT-V as a direct downstream effector of Sp1 signaling [20].

The Sp1 (specificity protein 1), a member of the Sp/KLF family, encodes a zinc finger transcription factor that binds GC-rich promoter regions [21]. It regulates a wide range of genes associated with cell proliferation, differentiation, apoptosis, carcinogenesis, stem cell maintenance, embryonic development, and cell tissue differentiation [22]. Sp1 is now recognized as a master regulator of cancer-related transcriptional programs [23].

Sp1 plays a key role in regulating the cell cycle and cancer stem cells (CSCs). In the G1 phase, nuclear Sp1 levels are elevated, promoting the expression of genes that enhance cell proliferation. In the M phase, Sp1 is phosphorylated, supporting mitotic processes. Sp1 is further phosphorylated by cyclin A-CDK complexes in the G2 phase, affecting its DNA-binding and chromatin condensation. By stimulating the expression of genes crucial for cell growth, Sp1 helps drive normal cell progression. However, dysregulation of Sp1 can lead to abnormal cell cycle progression and tumor development [24].

Sp1 is also modulated by multiple post-translational modifications, including phosphorylation, acetylation, glycosylation, and proteolytic processing, which regulate Sp1 activation [21]. Depending on its modification state, Sp1 can function as either a transcriptional activator or repressor [21]. Sp1 acts as an oncogenic factor that promotes uncontrolled cell proliferation, tumor survival, progression, and metastasis. Metastasis—the spread of cancer cells from one part of the body to another—is a significant cause of cancer-related mortality [25]. Elevated Sp1 levels have been reported in multiple cancers, including gastric cancer (GC), ovarian cancer, pancreatic cancer, hepatocellular carcinoma, glioblastoma, lung cancer, breast cancer, and colorectal cancer (CRC). In these malignancies, Sp1 expression is associated with poor clinical prognosis, partly due to its role in inhibiting apoptosis and promoting tumor progression [21,23].

#### 5.1.2. Background

Sp1 is a nuclear glycoprotein and transcription factor (TF) whose glycosylation occurs primarily through O-linked β-N-acetylglucosamine (O-GlcNAc) modification catalyzed by O-GlcNac transferase (OGT) in the nucleus (Figure 2). Increasing evidence suggests that Golgi-based glycosylation pathways can influence nuclear glycosylation dynamics.

β-1,4-GalT-V, a β-1,4-galactosyltransferase located in the Golgi, catalyzes the elongation of N- and O-linked glycans and is frequently upregulated in malignancies. Inhibition of β-1,4-GalT-V may alter intracellular pools of UDP-sugars—particularly UDP-Gal and UDP-GlcNAc—potentially affecting the hexosamine biosynthetic pathway and, subsequently, the O-GlcNAcylation of nuclear proteins.

Sp1 is among the most heavily O-GlcNAc-modified transcription factors (TFs). Its glycosylation modulates stability, transcriptional activity, and subnuclear localization. Therefore, β-1,4-GalT-V inhibition may indirectly impact Sp1 O-GlcNAcylation and its downstream transcriptional programs [26,27,28,29].

#### 5.1.3. Glycosylation-Dependent Regulation of Sp1: Linking Nuclear Signaling, Apoptosis, and Metastatic Potential

β-O-linked N-acetylglucosamine (O-GlcNAc) is a dynamic and abundant post-translational modification found on numerous cytosolic and nuclear proteins in metazoans. O-GlcNAc is dynamically processed by a unique set of enzymes that actively add and remove this modification. Functionally, O-GlcNAc regulates protein stability, subcellular localization, and protein–protein interactions [30]. Elevated O-GlcNAc levels have been linked to impaired insulin signaling and reduced activation of endothelial nitric oxide synthase through a pathway involving Akt inhibition [31,32]. Flux through the hexosamine biosynthetic pathway (HBP) modulates the activity of several TFs, including p53, Sp1, and NF-κB [33,34].

Sp1 and Sp3, which share DNA-binding motifs, collectively regulate more than 12,000 genomic binding sites [35], influencing genes involved in cell cycle regulation, angiogenesis, apoptosis, and genomic stability. SP1 is one of the best-characterized O-GlcNAc-modified proteins [36]. Multiple O-GlcNAc residues on Sp1 do not interfere with DNA-binding but reduce transactivation, as observed by Jackson et al. Later studies by Roos et al. showed that reduced Sp1 glycosylation led to its degradation, while glycosylated Sp1 peptides disrupted normal protein–protein interactions [26,27]. O-GlcNAcylation of Sp1’s transactivation domain inhibits its transcriptional activity [22]. Additionally, the glycosylated form of Sp1 appears to interact with components of the nuclear pore complex, possibly affecting its nuclear translocation [29].

There are at least 12,000 Sp1/3 binding sites in the human genome, linked to genes controlling nearly all cellular processes [36]. Sp1 target gene products include factors involved in cell cycle progression and arrest (e.g., cyclins), pro- and antiangiogenic factors associated with invasion and metastasis, and pro- and antiapoptotic factors involved in maintaining genomic stability [37,38,39].

FLIP protein (or CASP8 and FADD-like apoptosis regulator) modulates apoptosis and may bridge survival and cell death pathways in mammalian cells. Sp1 activates FLIP, whereas Sp3 represses its promoter. Thus, targeting Sp1- and Sp3-mediated FLIP regulation may offer a novel therapeutic strategy for cancers such as prostate cancer [40].

Most early studies suggested that Sp1 functions as an activator, while Sp3 acts as a repressor or, at most, a weak activator [41]. However, further research has shown that Sp1 and Sp3 can play dual roles in regulating gene expression [40,42,43]. Several studies have demonstrated that Sp3 is a more potent trans-activator of the p21 promoter than Sp1 [39]. Binding to specific Sp1/3 sites may determine whether Sp1 or Sp3 acts as an activator and may consequently enhance gene promoter function [40]. Sp1 and Sp3 have also been found to synergistically upregulate RAS association domain family 1A gene expression [44]. Notably, Sp1 and Sp3 exist in distinct complexes and do not co-occupy the Sp1/3 site in the upstream promoter region of the trefoil factor 1 gene in MCF-7 breast cancer cells [45].

Although Sp1 glycosylation is primarily mediated through nuclear O-GlcNAcylation by OGT (O-GlcNAc Transferase), growing evidence suggests that alterations in Golgi-based glycosylation pathways can influence nuclear glycosylation dynamics. β-1,4-GalT-V, a β-1,4-galactosyltransferase, catalyzes the elongation of O-linked glycans in the Golgi and is upregulated in several malignancies. Inhibition of β-1,4-GalT-V could shift intracellular UDP-sugar availability, particularly UDP-Gal and UDP-GlcNAc, which may alter flux through the hexosamine biosynthetic pathway and subsequently affect O-GlcNAcylation levels of nuclear proteins. As Sp1 is among the most heavily O-GlcNAc-modified TFs and its glycosylation state regulates its stability, transcriptional activity, and subnuclear localization, β-1,4-inhibition may indirectly disrupt Sp1 O-GlcNAcylation.

Such disruption could lead to Sp1 destabilization and reduced transcription of protumoral genes, including VEGF, CAIX, and FLIP (Figure 2). While β-1,4-GalT-V does not directly glycosylate Sp1, we speculate that its role in maintaining global glycosylation homeostasis may influence nuclear glycoprotein regulation through metabolic and signaling feedback. This suggests a novel axis in which targeting β-1,4-GalT-V could serve as a strategy to impair Sp1-driven transcriptional programs by modulating its glycosylation status. Future studies are needed to validate this indirect regulatory link and assess its therapeutic potential in Sp1-centric malignancies.

#### 5.1.4. Sp1 in the Tumor Microenvironment and Immune Modulation

Recent studies have highlighted the role of Sp1 within the tumor microenvironment (TME), particularly its Pro tumorigenic functions. Sp1 has been implicated as a key mediator of epigenetic programming and reprogramming in host cells infected with human papillomavirus (HPV). To counter this, plicamycin has been shown to inhibit Sp1 and concurrently enhance anti-PD-1 immunotherapy by reshaping the TME, suggesting Sp1 inhibition as a promising therapeutic strategy for HPV-related cancers [46]. HPV infection has also been linked to an increased risk of developing CRC, although the molecular mechanisms underlying this association remain under investigation [47].

Zhou et al. investigated the immunomodulatory role of Sp1 in GC, focusing on its effect on immune checkpoint inhibitors (ICIs), such as T-cell activation. Although cytotoxic T-cells can effectively destroy tumor cells, high expression of immune checkpoint receptors often suppresses this function. For example, overexpression of PD-L1 on tumor cells inhibits T-cell recognition and cytotoxicity by binding to PD-1 on T-cells, leading to T-cell inactivation and immune evasion. This suppression can be reversed through ICI therapy, which reactivates T-cell-mediated cytotoxicity and apoptosis, allowing the immune system to target cancer cells more effectively [48]. This approach harnesses the body’s natural antitumor immunity and may offer more effective treatments for cancers with high morbidity and mortality, such as CRC [48,49].

Sp1 has also been shown to interact with key transcriptional pathways involved in immune regulation. Notably, Nuclear Factor kappa-light-chain-enhancer of Activated B cells (NF-kB) and Sp1 both respond to various forms of cellular stress and external stimuli, playing essential roles in inflammation, apoptosis, and gene expression. Together, these transcription factors are pivotal mediators of biological processes relevant to tumor development and immune modulation [50].

Additionally, Wu et al. (2014) reported that fatty acid synthetase (FASN), a key cytosolic enzyme regulating de novo lipogenesis, in mammalian cells, is overexpressed in most cancers and contributes to cell survival and drug resistance [51]. FASN catalyzes the conversion of acetyl-CoA and malonyl-CoA into palmitate, a 16-carbon fatty acid. While FASN expression is low in normal cells due to sufficient dietary fat intake, it is upregulated in cancer cells to support rapid growth and membrane synthesis [51]. Interestingly, FASN overexpression has upregulated PARP-1, an enzyme involved in DNA damage response and repair, and increases non-homologous end joining (NHEJ) activity. FASN knockdown lessens PARP-1 expression and NHEJ repair regulation, thus suggesting that FASN has a role in maintaining the genome integrity against cancer cells [51].

Emerging studies show the potential of FASN as not only a lipid synthetic enzyme, but also a transcriptional regulator via Sp1. In CRC, FASN knockdown has been shown to increase Sp1 binding at the PLA2G4B promoter, altering downstream lipid signaling pathways that originate from lipids [52].

Zhou et al. further demonstrated, using genomic and single-cell omics analyses of gastric cancer patient samples, that Sp1 acts as a core regulator of cellular functions such as stability, proliferation, and stress response across diverse cell types [22].

#### 5.1.5. Sp1 and Low-Density Lipoprotein (LDL) in Atherosclerosis

Increased β-1,4-GalT-V activity and elevated LacCer levels have been detected in renal tubular cells from patients with the homozygous familial hypercholesterolemia [53], suggesting a role for Sp1 in atherosclerosis by modulating β-1,4-GalT-V expression and LacCer production.

During the early stages of atherosclerosis, elevated levels of low-density lipoprotein (LDL) within the blood promote its accumulation and oxidation (ox-LDL) within the arterial wall. This activates endothelial cells and upregulates adhesion molecules, enabling monocyte infiltration. These monocytes differentiate into macrophages that engulf ox-LDL and become lipid-laden foam cells. Foam cell accumulation contributes to plaque formation and initiates a chronic inflammatory response. Cytokines secreted by activated macrophages further recruit immune cells to the plaque and adventitia, exacerbating inflammation [54].

Sp1 modulates gene expression in endothelial cells exposed to ox-LDL, contributing to vascular injury and plaque progression [55]. As shown in Figure 3, this includes regulation of immune cell adhesion, foam cell formation, and proinflammatory signaling cascades.

#### 5.1.6. β-1,4-GalT-V, Clathrin-Coated Pits, and LDL Receptor Interactome

In tumor cells, β-1-,4-GalT-V localizes to clathrin-coated pits in the cell membrane, where it co-resides with LDL receptors, PD-L1, and other signaling molecules. In normal cells, β-1,4-GalT-V also localizes with the Trans-Golgi apparatus. These membrane microdomains serve as convergence points for receptor-mediated endocytosis and signal transduction. Interestingly, both the LDL receptor and β-1,4-GalT-V contain Sp1 response elements in their promoters. While native LDL downregulates β-1,4-GalT-V, oxidized LDL activates it by phosphorylation, highlighting the dual regulatory impact of extracellular lipid status on β-1,4-GalT-V mediated glycosylation.

This functional link between extracellular lipid environments and intracellular glycosylation reflects a feedback loop modulated by Sp1 [57].

#### 5.1.7. Cancer Stem Cells and the Sp1 Pathway

Cancer stem cells (CSC) are a subset of tumor cells with self-renewal and differentiation capacity. These cells resist standard therapies and often drive relapse and metastasis. A defining feature of CSCs is their ability to self-renew and differentiate into multiple cell types that contribute to tumor growth and progression [58]. In triple-negative breast cancer (TNBC), CSCs are associated with poor prognosis.

The type I transmembrane protein ε-Sarcoglycan (SGCE) was identified as a potential oncogene in TNBC, promoting CSC properties by stabilizing EGFR and facilitating the nuclear translocation of Sp1, which upregulates FGF-BP1—a secreted oncogenic protein. Depletion of SGCE partially reduces the breast CSC population by inhibiting FGF-BP1 transcription. Mechanistically, SGCE interacts with Sp1 to promote its nuclear translocation, enhancing FGF-BP1 expression. Secreted FGF-BP1 secretion then activates FGFR signaling, which in turn stimulates three dominant downstream pathways, RAS/MAPK, PI3K/AKT, and PLCγ, promoting CSC self-renewal and pluripotency [59].

The evidence supports Sp1 role as a transcriptional hub, regulating oncogenic gene expression and maintaining CSC traits. Sp1 upregulates oncogenic targets and suppresses tumor suppressor genes, contributing to the malignant potential in cancer-specific stem cells. Sp1 also influences cyclins and cyclin-dependent kinases (CDKs) thereby regulating the cell cycle, sustaining uncontrolled self-renewal, and driving tumor progression. The Sp1/FGF-BP1/FGF2/FGFR axis highlights how Sp1 governs the transcriptional pathways that promote cancer-specific stemness [60].

#### 5.1.8. Sp1 Interaction with PD-L1

PD-L1 (Programmed Death-Ligand 1), an immune checkpoint ligand, is upregulated in cancer cells to evade T-cell-mediated destruction [61]. Sp1 binds directly to the PD-L1 promoter and enhances its transcription. PD-L1 and Sp1 interactions are important for tumor immune evasion by reducing T-cell-mediated responses [61]. Given that PD-L1 is co-localized at the cell membrane and transcriptionally controlled by Sp1, regulating Sp1 activity can directly influence immune evasion by altering PD-L1 expression.

Tao et al. (2017) showed that a polymorphism (rs10815225) in the PD-L1 promoter [62] increases Sp1 binding affinity, leading to PD-L1 overexpression and increased gastric cancer risk [62]. These findings highlight how Sp1 contributes to immune evasion.

#### 5.1.9. Sp1 as a Therapeutic Target

Due to its role in regulating key cancer-related genes, SP1 has emerged as a therapeutic target in CRC. Antitranscriptional therapies, including DNA-binding agents and microRNAs—like miR-382—have shown efficacy in preclinical models. However, concerns remain due to Sp1’s role in housekeeping gene regulation and the limited availability of clinical data.

Sp1 target genes participate in cell proliferation and oncogenesis in many cancers, notably CRC, as their responsive genes are hallmarks of cancer [59]. Thus, targeting Sp1 has emerged as a strategy for CRC therapy. As binding TFs to specific DNA sequences in gene promoters is essential for transcriptional activation, DNA-binding drugs are potential agents that inhibit abnormal transcription in cancer. Many first-line cytotoxic agents have been clinically used to inhibit transcription. Sp1’s pro-oncogenic activity makes it a viable model for antitranscriptional therapies aimed at regulating transcriptional-level gene expression. However, targeting Sp1 can weaken CRC cells from chemotherapeutic drugs and induce cell death [63].

Recent studies show elevated Sp1 expression associated with lymph node metastasis, advanced TNM (Tumor, Node, Metastasis) stage, and poor prognosis in solid tumors including CRC. Anti-Sp1 approaches, such as transcriptional inhibitors or microRNA-based therapies targeting miR-382, a tumor suppressor gene, have demonstrated efficacy in preclinical models by suppressing tumor growth and metastasis. However, as Sp1 activity is linked with diverse genes, therapeutic targeting of Sp1 remains an issue due to concerns about such treatment’s effects on unintended targets. Additionally, the lack of clinical trial data limits arguments about long-term safety and efficiency. Targeting Sp1 offers insight into future CRC treatment, but further validation is required for practical clinical application [22,63].

#### 5.1.10. Sp1 and the Cell Cycle

Sp1 activity is modulated throughout cellular processes, such as cell cycle regulation, hormonal activation, apoptosis, and angiogenesis. Sp1-dependent transcription can be affected by changes in Sp1 abundance during the cell cycle, during which direct protein–protein interactions may not predictably involve direct binding to promoter DNA [54]. Sp1-dependent transcription also can be affected by changes in Sp1 abundance during the cell cycle [63]. Several posttranslational modifications, such as glycosylation, acetylation, phosphorylation, and SUMOylation, regulate Sp1 activity [64,65].

Because Sp1 activity is modulated throughout the cell cycle, it also is influenced by direct interaction with cyclins and CDKs. For example, CyclinE-CDK2 and CyclinA-CDK2 complexes phosphorylate Sp1, stabilizing it during key mitotic phases and forming stable complexes with this transcription factor [59]. Tapias et al. reported that CDK4, SKP2, Rad51, BRCA2, and p21 could interact with Sp1 and these interactions were confirmed by co-immunoprecipitation. CDK4, SKP2, Rad51, BRCA2, and p21 also activated the Sp1 promoter. E2F-DP1, Cyclin D1, Stat3, and Rb, among the Sp1 interacting proteins, activated the Sp1 promoter, whereas p53 and NFκB inhibited it. Additionally, SKP2, BRCA2, p21, E2F-DP1, Stat3, Rb, p53, and NFκB (Figure 3) had similar effects on an artificial promoter containing only Sp1 binding sites [66].

#### 5.1.11. Sp1 and the pH-Tome

Sp1 upregulates multiple genes critical to tumor metabolism and pH regulation, with six genes related to malignancy with a crucial role in the pH-tome, including HIF-1α (hypoxia-inducible factor 1 alpha), CAIX (carbonic anhydrase 9), NHE1 (sodium/hydrogen exchanger 1), NaV1.5 (an isoform of voltage-gated sodium channel), and V-ATPase protein which act as proton pumps. Sp1 is one of the promoters of HIF-1α, and these regulate the expression of pH-tome factors: NHE1, CAIX, Basigin (a chaperone of monocarboxylate transporters), a protein that forms part of proton pumps, and the NaV1.5 voltage-gated sodium channel. Thus, Sp1’s role in the pH-tome and its protumoral behavior make it a protein of interest [67].

Sp1 transcription factor also is a promoter of VEGF transcription after Ras activation and a promoter of human monocyte chemoattractant protein 1 (MCP-1) [68,69]. Sp1 interacts with TFs that regulate cell cycle and DNA synthesis, like cyclins, CDKs, DNA repair, and protein synthesis genes [70]. Included in its protumoral activities is the transactivation of cyclooxygenases-2 (COX-2) in gliomas, inducing interleukin-10 (IL-10) expression in regulatory T-cells (Tregs) and macrophages, and increasing transforming growth factor beta 1 (TGF-β1), as selected examples of its protumoral activities [71,72,73]. Given Sp1’s tumorigenic role, suppression of Sp1 has been shown to promote apoptosis in cancer cells [74].

#### 5.1.12. Role of SP1 in CRC Growth, Progression, and Metastasis

Sp1 has been shown to inhibit colon CSC proliferation and induce apoptosis in mice [56]. Sp1 suppresses key components of the autophagy and apoptosis pathways. It maintains cellular viability under stress conditions, making Sp1 key to cell survival and other biopharmaceutical processes where cell viability is critical [58].

Epigenetic changes are stable modifications in cellular function that do not alter the DNA sequence but regulate the expression of specific genes at the transcriptional level. DNA methylation is one such epigenetic mechanism; it controls gene expression by recruiting proteins involved in gene repression [56]. For example, when smokers quit, DNA methylation levels increase, leading to decreased gene expression, which raises the risk of cancer. In this context, abnormal expression of β-1,4-GalT-V has been linked to CRC [50]. β-1,4-GalT-V is transcriptionally regulated in cancer cells, as demonstrated by cloning the flanking region of the DNA promoter (the 5′ end where transcription starts), which allows transcription factors to bind. Sp1 is one of the TFs that bind to this promotor region, thereby playing a most prominent role in transcription activation [50].

In summary, Sp1 is a central transcriptional regulator linking many genes and proteins critical to health and disease and affecting metabolism, glycosylation, angiogenesis, immune evasion, and cancer stemness. Sp1 can be regulated by O-glycosylation and other post-transcriptional modifications, rendering new avenues for drug, antibody, and antibody–drug targeting to treat SP1 and β-1,4-GalT-V-centric diseases. Below, we present evidence showing how Sp1 regulates β-1,4-GalT-V, Notch-1, and other β-1,4-V-centric interactomes.

### 5.2. β-1,4-GalT-V Interaction with Notch-1

In endothelial cells, Notch-1 is glycosylated by β-1,4-GalT-V at its extracellular domain (ECD). This modification influences ligand binding, proteolytic cleavage, and translocation of Notch-1 responsive genes. siRNA-mediated knockdown of β-1,4-GalT-V reduces Notch-1 glycosylation and downstream angiogenic signaling. Additionally, Notch-1 has an intracellular domain and an extracellular domain. Glycosylation is achieved by adding two O-linked sugars. This allows a cascade of activities in which the NICD separates from the NECD (with the help of ADAM family proteases), with both serving different functions. The NECD interacts with the Notch ligands—DLL and Jagged. Galectin-3 is also involved in this process (Figure 4).

#### 5.2.1. β-1,4-GalT-V Regulates Notch-1 Functions

##### Background

Notch signaling plays a central role in cancer biology, governing processes such as proliferation, apoptosis, angiogenesis, and immune evasion. β-1,4-GalT-V has been shown to glycosylate Notch-1, a transmembrane receptor involved in these pathways, thereby modifying its activity.

This glycosylation occurs within the epidermal growth factor (EGF)-like repeats of Notch-1’s extracellular domain (ECD), which contains serine and threonine residues that serve, as Notch signaling plays a central role in cancer biology, governing processes such as proliferation, apoptosis, angiogenesis, and evasion and tumor neovascularization [75]. β-1,4-GalT-V has been shown to glycosylate glycosylation sites. The O-linked glycans in this region affect ligand binding, proteolytic cleavage, and intracellular domain (NICD) translocation. These processes are essential for transcriptional activation of Notch targeted genes.

Notch-1 is a glycoprotein consisting of an intracellular and extracellular domain (NICD and NECD, respectively). Notch-1 has 36 EGF-like repeats influencing Notch functionality, and an LIN12-Notch repeat area, with a heterodimerization domain (Figure 4). β-1,4-GalT-V and β-1,4-GalT-IV catalyze the addition of UDP-galactose to GlcNAcβ1-6Man and the GlcNAcβ1-4 Man, respectively [8]. Wang et al. [8] showed that O-galactosylation can affect Notch signaling in cancer by altering ligand binding and receptor behavior. Thus, it targets TFs like E1AF (a transcription factor important for cancer metastasis) and Sp1 for their ability to metastasize cells [8]. Notch signaling is a highly conserved cell–cell communication pathway essential for various developmental and physiological processes. The Notch family comprises four transmembrane receptors (Notch-1, Notch-2, Notch-3, and Notch-4) interacting with membrane-bound ligands on neighboring cells [8]. This interaction triggers a signaling cascade within the cell, influencing gene expression, and ultimately determining cell fate, proliferation, and differentiation. Notch-1 is a prominent member of this family, playing a crucial role in numerous developmental processes [8]. It participates in cell fate determination in various tissues, including the nervous system, heart, and blood vessels. Additionally, Notch-1 signaling regulates cell proliferation and differentiation in adult tissues, contributing to tissue homeostasis and regeneration. For further details, the readers are referred to an excellent review on the Notch family of proteins [8]. This review focuses primarily on β-1,4-GalT-V interactions with Notch-1 and how it functions in various regulating, modulating, and other actions.

N-glycosylation of proteins often regulates cell behavior. For example, the use of an N-glycosylation inhibitor, tunicamycin, has been shown to block tumorigenesis and the self-renewal capacity of glioma stem cells [76]. Tunicamycin also inhibited LDL receptors in cultured human skin fibroblasts and, consequently, cholesterol metabolism [77]. Previous studies have shown that in human endothelial cells, VEGF plays a critical role in angiogenesis/tube formation and that treatment with antiangiogenics—for example, Avastin, an antibody against VEGF—can markedly reduce tumor growth. However, subsequently, these same tumor cells can actively engage in tumorigenesis [19,78]. We observed that treating human arterial endothelial cells with D-PDMP, an inhibitor of β-1,4-GalT-V or β-1,4GalT-V gene ablation using GalT-V-siRNA, completely blocked VEGF-induced angiogenesis [79]. Moreover, treatment with LacCer in these β-1,4-GalT-V-deficient endothelial cells resumed angiogenesis. Thus, multiple lines of evidence suggest the existence of VEGF-independent angiogenesis governed by β-1,4-GalT-V. However, how Notch-1 participates in this VEGF-independent pathway leading to angiogenesis remains to be determined.

Subsequent studies using glioma stem-like cells revealed that β-1,4-GalT-V gene ablation in these cells, using β-1,4-GalT-V shRNA, mitigated their trans-differentiation into endothelial cells. This is accompanied by a marked reduction in PECAM-1/CD31, an integral protein in endothelial cell junctions, and tube formation [80]. Additionally, when these β-1,4-GalT-V-deficient glioma cells were tagged with green fluorescent protein (GFP) and implanted in the frontal lobe in nude mice, it significantly increased the survival of tumor-bearing mice compared with mice inoculated with glioma stem cells expressing β-1,4-GalT-V. These findings showed that β-1,4-GalT-V KO can inhibit the trans-differentiation of glioma stem cells into endothelial cells in vitro and in vivo [80].

Previous results have suggested that the catalytic domain in β-1,4-GalT-V contains two conservative domains at Tyr 268 and Trp294 that are required to engage in its ability to glycosylate [20]. To examine the role of these resides on glioma genesis, shRNA-resistant, hemagglutinin-tagged point mutants of β-1,4-GalT-V at Tyr 268 and Trp 294 were used in this study [20].

Inoculation of mice revealed that glioma Gensis was achieved only with the wild-type β-1,4-GalT-V-positive cells but not with the β-1,4 GalT-V mutant cells. Thus, the galactosylation activity of β-1,4-GalT-V regulates the trans-differentiation of glioma-like cells into endothelial cells.

Notch-1 is an N-glycosylated protein and is required in the regulation of endothelial trans-differentiation of glioma stem-like cells into endothelial cells [81]. To determine if β -1,4-GalT-V regulates Notch-1 signaling to trans-differentiate glioma-like stem cells to endothelial cells, the level of Notch-1 was measured in β-1,4-GalT-V mutant cells. These studies revealed a decreased level of Notch-1 and NICD protein in the mutant cell. Further, mutant cells reacted weakly to a lectin called *Riccinus communis*, which specifically binds to galactose residues compared to wild-type glioma cells [81], and β-1,4-GalT-V could be immunoprecipitated with Notch-1. Thus, β-1,4-GalT-V galactosylates Notch-1. As glycosylation of proteins can facilitate the transmigration of proteins to the cell surface, subsequent studies also showed that b-1,4-GalT-V mutant cells had less Notch-1 located on the cell surface as compared to wild-type glioma cells, thus blocking the mutant cells from interacting with galectin 3 [82]. Collectively, these studies reveal that β-1,4-GalT-V can regulate a VEGF-independent pathway to induce angiogenesis via recruiting LacCer and Notch-1 in addition to the trans-differentiation of glioma stem-like cells into endothelial cells [80].

By mediating the galactosylation of Notch-1, β-1,4-GalT-V enhances Notch-1’s stability and promotes its localization to the cell surface, which is crucial for effective Notch signaling. The interaction of β-1,4-GalT-V with Notch-1 facilitates its signaling functions, which are essential for cell proliferation and metastasis in different types of cancers, thereby linking aberrant glycosylation events with aggressive tumor behavior, particularly in glioma and breast cancer contexts. It is important to note that another function of Notch-1—binding to Galectin-3—is also affected when Notch-1 galactosylation is inhibited [5].

Importantly, β-1,4-GalT-V’s impact on Notch-1 also extends to T-cell differentiation, highlighting its pivotal role in immune responses. This suggests that modulation of b-1,4-GalT-V could serve as a potential therapeutic strategy in targeting Notch-1-related signaling pathways in cancer treatment [5]. Notch-1 has previously been targeted for colorectal cancer treatment but not in relation to β-1,4-GalT-V. Withaferin-A inhibited the Akt, NF-kB, and mTOR pathways, triggering apoptosis in colon cancer cells without adversely affecting normal colonic cells [83].

#### 5.2.2. Relationship Between Notch-1 and T-Cell Development

β-1,4-GalT-V influences Notch-1 localization to the cell membrane [5]. Losing β-1,4 GalT-V leads to reduced Notch-1 cell-surface expression and less effective signaling [5]. This also disrupts interactions with ligands like DLL-4. This is integral to the development of T-cell progenitors [5].

Bone marrow progenitors travel to the thymus, wherein Notch-1 signaling activates, differentiating these cells into T-cells. If Notch-1 signaling is blocked or inactivated, the T-cell development fails [58]. Hes1 and Hes4 (Figure 4) are critical for early T-cell development. DLL1 and DLL4 are important Notch1 binding ligands. In the thymus, thymic epithelial cells express DLL4 and interact with Notch1 on hematopoietic progenitors. When thymocytes progress into CD4+CD8+ cells, DLL4 expression decreases. Pre-T-cell receptor (pre-TCR) signaling is important for β-selection. Notch interacts with those pre-TCR signals to promote proper T-cell development [58]. GATA3, which is a key transcription factor for T-cell maturation and Th1/Th2 lineage choice, is a direct Notch-1 target gene. Thus, this pathway is dependent on β-1,4-GalT-V [58].

A study by Asano et al. highlighted how these two pathways, namely, Notch-1 and TGF-β signaling pathways, interact [82,84]. Notch-1 has been shown to enhance TGF-β (Transforming Growth Factor Beta)-mediated effector functions. These are key in regulatory T-cell (Treg) function [85]. When Notch-1 signaling is inhibited, the suppressor function of Treg is inhibited. Treg is important for regulating the immune system by way of preventing autoimmune diseases and maintaining tolerance to antigens created by the body. However, inhibition of Treg can lead to cells being hidden from the immune system—this enables cancer cells to continue to grow and proliferate. It has also been found that SMAD3, which is a protein that plays a key role in transmitting chemical signals from the cell surface to the nucleus, is also key in this interaction. When SMAD3 is phosphorylated (after being nuclear translocated), it interacts as part of the TGF-β pathway to send signals from the plasma membrane to the nucleus.

Thus, β-1,4-GalT-V enhances Notch-1 stability and function through galactosylation, linking it to cancer progression, immune regulation, and angiogenesis. Its impact on T-cell differentiation and tumor behavior highlights its broad biological relevance. Disruptions to this pathway affect the ability of Notch-1 to localize to the cell surface and affect its signaling ability. This further indicates how β-1,4-GalT-V could potentially be a focal tool in learning more about cancers that have abnormal Notch-1 activity as well as offering a potential clinical strategy.

### 5.3. β-1,4-GalT-V Interaction with Wnt

Wnt-1 is a member of the Wnt family of secreted glycoproteins that regulate numerous cellular processes, including proliferation, differentiation, and embryonic development [86,87]. Specifically, Wnt-1 plays a key role in the formation of the central nervous system, neural tube closure, midbrain and cerebellum patterning, and differentiation of dopaminergic neurons [88].

Wnt signaling is initiated through interaction with a family of highly conserved receptors known as Frizzled (FZD). Together, Wnt and FZD regulate two major signaling branches: the canonical (β-catenin-dependent) and non-canonical (β-catenin-independent) pathways [85,88]. In the canonical pathway, Wnt binding stabilizes β-catenin, allowing its translocation into the nucleus where it activates transcription of target genes via T-cell factor/lymphoid enhancer-binding factor (TCF/LEF) transcription factors. This pathway plays essential roles in embryonic development, stem cell proliferation, tissue homeostasis, and cell growth, and its dysregulation has been implicated in tumor progression across several cancers, including breast cancer, colorectal cancer, and melanoma [86,89]. In contrast, the non-canonical Wnt pathways function independently of β-catenin and TCF/LEF, and are subdivided into the Wnt/Ca^2+^ and Wnt/planar cell polarity (PCP) pathways, as shown in the figure. The Wnt/Ca^2+^ pathway modulates intracellular calcium signaling via release from the endoplasmic reticulum, while the Wnt/PCP pathway regulates cell polarity, division, and migration through cytoskeletal rearrangement [90,91]. While Wnt signaling supports stem cell maintenance and self-renewal under normal physiological conditions, its aberrant activation disrupts these processes and promotes tumorigenesis.

#### 5.3.1. The Canonical Wnt/FZD Signaling Pathway

The canonical Wnt/β-catenin signaling pathway, a highly conserved pathway in biological evolution from lower *Drosophila* to higher mammals, regulates β-catenin stability and is dependent on β-catenin gene expression. As the degradation complex is inhibited, β-catenin is no longer degraded and accumulates in the cytoplasm. It is then transferred to the nucleus and participates in the transcription of relevant target genes together with LEF/TCF TFs [92].

In the absence of extracellular Wnt ligands, the signaling switch is not activated, the canonical pathway is disabled, and the central effector β-catenin is continuously phosphorylated (Figure 5) and degraded, preventing abnormal cellular responses caused by the canonical pathway [93].

#### 5.3.2. Non-Canonical Wnt/FZD Signaling

The non-canonical Wnt pathway consists of Wnt ligands Wnt4, Wnt5a, Wnt5b, Wnt6, Wnt7a, and Wnt11. When these ligands bind to cell surface FZD receptors and co-receptors, the DVL-dependent effector proteins or Ca^2+^-dependent signaling cascade are activated, triggering intracellular signal transduction.

The Wnt/Ca^2+^ pathway is centered on elevated intracellular Ca^2+^ (Figure 5) and activates signaling factors such as PKC, CaMKII, CaN, NFAT, and TFs to exert cell adhesion and gene expression [94].

Non-canonical Wnt signaling can contribute to the development of inflammation and diseases such as cancer in human subjects by affecting cell motility, polarity, and migration [95].

#### 5.3.3. Relationship Between β-1,4-GalT-V and Wnt-1

In the development and homeostasis of cells, canonical Wnt signaling is induced when Wnt ligands bind to their specific cell surface. β-catenin, a key effector of the Wnt pathway, forms transcriptional complexes with TCF family transcription factors, activating genes involved in cell proliferation. This process is often upregulated in cancer, frequently due to stabilizing mutations in β-catenin or loss-of-function mutations in the tumor suppressor gene APC (adenomatous polyposis coli). We speculate that β-1,4-GalT-V can glycosylate precursor proteins in the Wnt/β-catenin pathway [96], thereby influencing Wnt ligand activation and receptor function. This glycosylation can modulate downstream effects on differentiation, tissue remodeling, infiltrative growth, and metastasis in colorectal and breast cancers.

#### 5.3.4. Relationship Between Wnt-1 and T-Cells

Wnt signaling also plays an essential role in hematopoiesis and T lymphocyte development in the thymus. When dysregulated, the pathway contributes to tumor progression by shaping the tumor microenvironment (TME). TCF/LEF transcription factors are principal nuclear effectors of Wnt signaling, despite the existence of other β-catenin-binding transcription factors [97].

Studies have shown that Wnt signaling supports the generation of CD8+ memory stem T-cells with enhanced self-renewal and long-term persistence capabilities. For example, CD8+ memory stem cells are key to sustained immune protection and can rapidly respond to infections upon repeated exposure. Also, Wnt3a exposure halts CD8+ T-cell differentiation into effector cells, preserving an undifferentiated population with strong antitumor potential [86]. These memory stem cells outperform effector memory T-cell subsets in durability and responsiveness. This finding has implications for vaccine development and immunotherapy [98].

#### 5.3.5. Relationship between Wnt-1 and Notch-1

Studies show that Notch and Wnt/β-catenin signaling are interlinked in stem and progenitor cells, with each able to regulate the other’s activity. The outcome of signaling through either pathway depends on the cellular differentiation stage. In one study, overexpression of the Notch1 intracellular domain (NICD1) in HCT-116 colorectal cancer cells—cells with naturally low NICD1 expression—led to enhanced nuclear translocation of β-catenin [99]. Furthermore, immunohistochemical analysis of 189 colon cancer tissue samples showed no co-localization of NICD1 and β-catenin in the nucleus. Overexpression of NICD1 increased Wnt pathway activity (measured via luciferase reporter assay), whereas NICD1 knockdown reduced this activity [100]. These results suggest that NOTCH1 activation promotes β-catenin nuclear localization and Wnt pathway upregulation.

#### 5.3.6. Wnt-1 and Colorectal Cancer

The Wnt signaling pathway plays an important role in colorectal cancer (CRC) survival, proliferation, and self-renewal. However, most of these factors are dysregulated in cancer stem cells. The canonical Wnt pathway is involved in the rapid cycling of cancer stem cells, while the non-canonical pathway maintains the slow cycling of cancer stem cells via TGFβ signaling cascades [86]. SOX9, a transcriptional activator, enhances CRC growth through a positive feedback loop with prominin 1 (PROM1) [86]. Intestinal stem cells act as a homeostatic balance but promote cancer stem cells in intestinal tumorigenesis when Β-catenin in intestinal stem cells is activated, stimulating the development of CRC due to inhibition of the adenomatous polyposis coli (APC), a tumor suppressor gene [86].

As metastasis in CRC can be lethal and the dysregulation of the β-catenin pathway is involved in CRC metastasis, it is important to understand how the Wnt signaling pathway affects metastasis. In the nucleus, β-catenin and TCF/LEF activate target genes like Twist and Slug, which regulate epithelial–mesenchymal transition (EMT), a critical process for metastasis [88]. Additionally, other proteins and genes, such as RNF128, SMYD2, and FOXO4, modulate Wnt/β-catenin signaling and affect CRC proliferation and invasion [86]. Particularly, the overexpression of SMYD2 enhances metastatic ability by suppressing adenomatous polyposis coli 2 (APC2), a Wnt/β-catenin inhibitor. Similarly, FOXO4 inhibits CRC cell metastasis through the APC2/β-catenin axis, which demonstrates the protein’s potential as a therapeutic target [100].

Metabolic reprogramming is also essential for CRC metastasis, enabling cancer cells to adapt to different conditions in other organs. The non-canonical Wnt/PCP pathway is dysregulated in CRC, which promotes cell motility and invasiveness. Proteins Wnt11 and FZD7 enhance CRC proliferation and invasion through JNK pathways (Figure 5).

#### 5.3.7. Interaction Between β-1,4-GalT-V, Wnt-1, and Colorectal Cancer

While the Wnt/PCP pathway is involved with colorectal cancer, β-1,4-GalT-V influences colorectal cancer through the Wnt/Ca^2+^ signaling pathway [100]. β-1,4-GalT-V alters the function of Wnt-1 ligands through glycosylation, affecting the ability of Wnt-1 to bind to Frizzled and ROR2 receptors. This alteration could increase or inhibit the downstream signaling cascade involving IP_3_, DAG, and calcium release, which can impact TFs such as NFAT. In colorectal cancer, β-1,4-GalT-V is shown to be overexpressed and has been associated with increased tumor cell migration and metastasis. Through interaction with the Wnt/Ca^2+^ signaling pathway, we speculate that β-1,4-GalT-V may enhance the role of the Wnt pathway in actin cytoskeleton remodeling and cell migration, which may contribute to cancer growth (Figure 5). Understanding how these pathways contribute to epithelial–mesenchymal transition (EMT) and metastasis is crucial for developing new therapeutic strategies against CRC.

#### 5.3.8. Therapeutic Approaches Targeting Wnt-1

Targeting the Wnt/β-catenin signaling pathway represents a therapeutic approach in the treatment of colorectal cancer [101]. Among the pathway’s regulatory components, porcupine (PORCN), an O-acyltransferase essential for the secretion of Wnt ligands, is a promising therapeutic treatment due to its upstream control of pathway activation [102].

Recent studies show that a PORCN inhibitor, WNT974 (LGK974), has advanced into clinical trials. WNT974, as discussed in Table 1, was evaluated in treating melanoma, breast cancer, CRC, and pancreatic adenocarcinoma. In Phase I, 94 patients were given WNT974 at doses of 5–30 mg daily depending on the dosing schedules [101]. Out of the 94 patients, skin biopsies of 52 patients and 35 paired tumor biopsies showed reduction in axis inhibition protein 2 (AXIN2), which confirmed the inhibition of the Wnt pathway [101]. AXIN2 is a negative regulator of the Wnt/β-catenin signaling pathway. However, in Phase Ib/II of the clinical studies, 15 out 20 patients reported frequent bone fractures and hypercalcemia after being given WNT974, encorafenib, and cetuximab. Due to these adverse effects, the study was discontinued [102]. Though this study did not show desired results, new treatments that inhibit the Wnt/β-catenin signaling pathway are being evaluated and may shed new light on alternative treatments for CRC.

### 5.4. β-1,4-GalT-V Interaction with Frizzled

#### 5.4.1. Background

Frizzled, or FZDs, belong to the F class of the superfamily of the human G-coupled protein receptors and are activated by the Wingless/Int-1 (Wnt) family [94]. Belonging to a large GPCR family, ten FZDs (FZD1-10) constitute a conserved family of receptors, ranging from 500 to 800 amino acids in length. The primary molecule that binds to these ten receptors is the Wnt family protein [103].

Disheveled (DVL), a crucial protein, transduces signals to the three Wnt/FZD pathways by acting as an intracellular hub protein binding to the FZD and forming a signaling complex called a “signalosome” (Figure 5), leading to downstream effects [104].

The FZD pathway is pivotal in maintaining embryonic tissue development and homeostasis of adult tissue. However, if not regulated, they promote cancer development via multiple channels and mechanisms. Wnt binds to individual FZDs and their co-receptors to activate intracellular signaling, which triggers modifications in cell morphology and motility [105]. The intracellular signaling system consists of three principal streams—one canonical Wnt/B-catenin pathway [92] and two non-canonical Wnt/B-catenin pathways consisting of a Wnt/planar cell polarity (PCP) pathway [92] and a Wnt/calcium signaling pathway [92], which are explained in the previous section.

#### 5.4.2. Frizzled Receptors Orchestrate Canonical and Non-Canonical Wnt Signaling in Colorectal Cancer Progression

FZD_2_ and its ligands (Wnt 5a and Wnt 5b), are reported to be overexpressed in colon cancers [106]. Kirikoshi et al. demonstrated that among normal human tissues, small amounts of FZD4 mRNA occur in the colon [106]. Mutations of *APC* and *CTNNB1* (gene encoding β-catenin) in the Wnt pathway play a crucial role in tumorigenesis [107,108]. FZD_7_ activates β-catenin-dependent Wnt pathway signaling despite APC or CTNBB1 mutations existing downstream [109]. FZD_7_ promotes tumor growth by invoking epithelial characteristics, which reduces the potential of cells to disperse [110]. This causes CRC cells to remain cohesive and promotes local tumor growth. In addition to its role in the β-catenin-dependent Wnt pathway, FZD_7_ may also transmit signals via β-catenin-independent pathways in CRC. FZD_7_ knockdown decreased c-Jun, p-JNK, and p-c-Jun protein levels and RhoA activation, which are indicators of the Wnt–PCP pathway [111]. FZD_10_ gene expression was found to be upregulated in CRC once polyps were formed and progressed to CRC. The study also indicated that cancer cells that were immune-positive for FZD_10_ displayed less nuclear accumulation of β-catenin, suggesting that FZD_10_ functions via the non-canonical pathway [112].

#### 5.4.3. β-1,4-GalT-V Promotes Breast Cancer Stemness and Tumor Progression

β-1,4-GalT-V was found to significantly increase 1A1 (ALDH1A1), which is a stem cell marker highly correlated with BCSC markers, resulting in poor prognosis [113]. This resulted in production of CD44^+^CD24^−/low^ cells (a molecular marker for BCSCs) and overexpression of β-1,4-GalT-V, resulting in dramatic tumor growth in vivo. β-1,4-GalTs (except β-1,4-GalT-VI) were upregulated in aggressive breast carcinomas in comparison with mammary tissues [114]. However, only β-1,4-GalT-V was found to be positively associated with the BCSC marker ALDH1A1 [113].

In vitro studies conducted using MCF-7 and MCF-7ADR (Adriamycin-resistant MCF-7) breast cancer cell lines indicated that β-1,4-GalT-V was highly expressed in MCF-7 mammospheres. β-1,4-GalT-V overexpression (after 48 h of plasmid transfection) in these cells significantly increased ALDH1A1 expression, with a corresponding increase in ALDH activity in MCF7-ADR cells. This confirms that β-1,4-GalT-V plays a role in the formation of BCSC stemness. Knockdown of β-1,4-GalT-V (after 48 h of plasmid transfection) resulted in a decline in CD44^+^CD24^−/low^ cells. However, a dose-dependent decrease in β-1,4-GalT-V was not reported. In vivo studies determined that mice injected with MCF-7 ADR with a higher expression of β-1,4-GalT-V exhibited a dramatic increase in tumor growth [113].

β-1,4-GalT-V protects FZD_1_ from degradation via the lysosomal pathway, promoting the Wnt/β-catenin pathway in BCSCs, and we speculate that β-1,4-GalT-V does this via N-linked glycosylation. β-1,4-GalT-V located on the cell surface of breast carcinoma cells did not play a role in the stemness of BCSCs [113]. This indicates that β-1,4-GalT-V displays CSC-related specificity and might play an important role in breast cancer.

#### 5.4.4. FZD-Based Targeted Therapy

Table 1 highlights therapeutic approaches which target the FZD pathway; these are discussed below.

Treatment using Vantictumab (OMP-18R5), a human monoclonal antibody targeting FZD_1/2/5/7/8_, prevents phosphorylation of the coreceptor LRP [114].

In a Phase 1b trial for a period of 4 weeks involving 48 patients with locally recurrent or metastatic HER2-negative breast cancer, Vantictumab (3.5–14 mg/kg on days 1 and 15 or 3–8 mg/kg on day 1 of every 28-day cycle) combined with weekly paclitaxel (90 mg/m^2^ on days 1, 8, and 15) achieved an overall response rate (ORR) of 31.3% and a clinical benefit rate (CBR) of 68.8%. Notably, a six-gene Wnt pathway signature correlated with improved progression-free and overall survival. However, six patients experienced bone fractures, which, despite occurring outside the dose-limiting toxicity window, raised concerns about bone safety and limited further clinical development of Vantictumab in this setting [115].

Ipafricept (IPA; OMP-54F28) is a recombinant protein targeting FZD8 using the extracellular Wnt binding region of the human FZD8 receptor. OMP-54F28 inhibits the growth of many types of tumors, and Phase 1b clinical trials demonstrated the therapeutic benefits of this inhibitor in combination with other drugs [116]. Ipafricept was evaluated in a Phase 1b trial for recurrent platinum-sensitive ovarian cancer up to six 21-day cycles given on a q3-weekly (every 3 weeks) schedule in combination with paclitaxel and carboplatin. Specifically, Ipafricept was given on day 1 and chemotherapy on day 3 of each cycle. Among 37 treated patients, the ORR (observed response rate) was 75.7%, with a median progression-free survival of 10.3 months and overall survival of 33 months. Despite these encouraging outcomes, dose-dependent bone fragility fractures were observed, leading to the discontinuation of development in this indication [117].

Thus, FZD receptors play a pivotal role in Wnt signaling and are linked with the development of cancer. β-1,4-GalT-V protects FZD_1_ from degradation, thus promoting stemness and proliferation of cancer cells. Understanding this link is crucial because β-1,4-GalT-V figures prominently in tumorigenesis and the inhibition of the Frizzled pathway. Blocking β-1,4-GalT-V may lead to the development of novel therapeutic approaches.

### 5.5. Hedgehog Pathway

Hedgehog (Hh) signaling plays a vital role in embryogenesis and somatic development, and its abnormal activation has been linked to various human cancers [118,119]. This pathway is crucial for the growth and differentiation of multiple tissues, including the gastrointestinal tract [120,121]. In mammals, Hh signaling is initiated when one of three ligands binds to the transmembrane receptor Patched 1 (Ptc1) [122]; the ligands include Indian Hedgehog (Ihh), which is critical for chondrocyte differentiation [123]; Sonic Hedgehog (Shh), which regulates embryonic and post-natal developmental processes [124]; and Desert Hedgehog (Dhh), a key regulator of gonadal tissue development [124]. This binding relieves the inhibition of the transmembrane protein Smoothened (Smo), enabling its activation. Smo, located on the primary cilium, is a member of the seven-transmembrane receptor family and is structurally similar to the Frizzled family [120].

Shh and Ihh (Figure 6) are expressed in gastrointestinal tracts, whereas Dhh expression is restricted to the nervous system and testes [125]. The transduction response is via Ptc and Smo, which are transmembrane proteins that activate downstream TFs of the Gli family (Gli1, Gli2, and Gli3). Smo subsequently activates the Gli TFs [126].

Hh activation occurs via (1) an Hh ligand-dependent activation or (2) a ligand-independent activation. The Hh ligands bind to Ptc, activating the G pro tein-coupled receptor-like transmembrane protein, Smo [127]. Ptc inhibits Smo via an unknown mechanism when Hh is absent [128]. Smo signals intracellularly to mediate the three Gli zinc finger TFs.

Smo then activates two different intracellular signaling cascades [129]:(1)A non-canonical, ligand-independent pathway;(2)A canonical, ligand-dependent pathway.

Smo-regulated canonical signaling pathway involves activation of Gli2, where Gli2 resides in the cytoplasm and is linked to a suppressor complex composed of Fused kinase (Fu), Suppressor of Fused (SuFu), and Costal2 [130]. Gli1, Ptc, and Hhip are general transcriptional targets of the canonical Hh signaling activity [131]. In the absence of Ptc ligand, Smo remains inactive, thereby inhibiting the transcription of Gli1, and the release of Gli2 and Gli3 is cleaved to generate repressor isoforms (Gli3Rs). When Smo is activated by Ptc, Gli2 is released, Gli3 repressor function is inhibited, and Gli1 is transcriptionally active. This leads to transcription of Gli1 and Gli2 target genes (Figure 6) [132].

Several studies indicate that the Hedgehog (Hh) signaling pathway contributes to colorectal cancer (CRC) progression by regulating colonic enterocyte differentiation and promoting tumor cell survival, metastasis, and proliferation through elevated Hh-Smo-Gli activity [132]. Both Sonic Hedgehog (Shh) and Indian Hedgehog (Ihh) [133] homologs play distinct roles in CRC development. Shh primarily promotes tumorigenesis, while Ihh appears to exert inhibitory effects [134]. Shh enhances angiogenesis, cell proliferation, and metastasis, suppresses tumor suppressor genes, and exhibits overexpression positively correlated with cancer stem cells (CSCs), CRC cells, and tumor masses [135,136]. In contrast, Ihh downregulation has been identified as an early event during CRC initiation [137].

Among the downstream components, Smoothened (Smo) plays a critical role in modulating Hh activity in CRC [138]. Additionally, crosstalk between Hh/Gli1 and Wnt/β-catenin pathways has been observed, in which their simultaneous activation leads to the upregulation of oncogenes such as cyclin D and c-Myc, contributing to tumorigenesis in gastrointestinal malignancies including CRC [139,140].

#### 5.5.1. β-1,4-GalT-V and Other Glycotransferases in Hedgehog Signaling Pathways

Zhou et al. demonstrated that β-1,4-galactosyltransferase I (β-1,4-GalT-I) plays a critical role in maintaining Hedgehog (Hh) signaling and chemoresistance in chronic myeloid leukemia cells. In Adriamycin-resistant K562 (K562/ADR) cells, β-1,4-GalT-I expression was significantly elevated compared to the parental K562 line. Silencing β-1,4-GalT-I using siRNA (transfection duration: 24 h) led to a marked reduction in the expression of key Hh pathway components, including Shh, smoothened (Smo), Patched (Ptch1), and Gli1. This inhibition of Hh signaling was accompanied by increased sensitivity of the resistant cells to chemotherapeutic agents, indicating that β-1,4-GalT-I contributes to drug resistance through modulation of Hh pathway activity. These findings highlight a novel functional link between glycosylation and oncogenic signaling and suggest that β-1,4GalT-I may serve as a potential therapeutic target for overcoming drug resistance in leukemia. This resulted in an increased sensitivity to chemotherapeutic drugs [141].

In another study, Zhou et al. [142] demonstrated that overexpression of β-1,4-GalT-I and β-1,4-GalT-V impacts the development of multidrug resistance of adriamycin-resistant human leukemia cell lines, which is characteristic of tumor cells, and subsequent suppression of either gene restored chemosensitivity in this cell line. The study demonstrated that β-1,4-GalT-I and β-1,4-GalT-V were increased in these drug-resistant leukemia cell lines at both gene and protein levels in comparison to drug-resistant parental controls. Suppression of β-1,4-GalT-I using shRNA (after a screening period of 60 days) in HL60/ADR cell lines decreased cell proliferation by a therapeutic drug, or suppression of β-1,4-GalT-V using shRNA (after a screening period of 60 days) restored chemosensitivity [142].

To determine the effects of knockdown of β-1,4-GalT-I or β-1,4-GalT-V gene on chemosensitivity, nude mice bearing HL60/ADR, HL60/ADR-β-1,4-GalT-I shRNA and HL60/ADR-β-1,4-GalT-V shRNA xenografts were used. In the HL60/ADR-control shRNA group, there was no significant difference in tumor volume between the mice groups with and without drug treatment. However, for mice in the HL60//ADR β-1,4-GalT-I shRNA group and HL60//ADR-β-1,4-GalT-V shRNA group, tumor volumes were significantly decreased with drug treatment in comparison to those without drug administration [142].

Mechanistically, knockdown of either GalT isoform reduced expression of key drug efflux proteins, including P-glycoprotein (P-gp) and multidrug-resistance-associated protein 1 (MRP1). Conversely, overexpression of β-1,4-GalT-I and β-1,4-GalT-V elevated P-gp, MRP1, and Hh signaling proteins (Shh, Smo, Gli1). Importantly, clinical samples from 37 of 71 CML and 29 of 75 AML patients revealed elevated P-gp and MRP1 in drug-resistant leukemia, with correspondingly higher β-1,4-GalT-I and β-1,4-GalT-V levels, supporting their role as clinical markers and potential therapeutic targets [142]. Higher levels of β-1,4-GalT-I, and β-1,4-GalT-V were found in chemoresistant leukemia samples, but expression levels of other members of the β-1,4-GalT-V family showed no difference between the drug-resistant group and chemosensitive group [143].

Thus, inhibition of the Hh pathway may act as an effective therapeutic treatment for overcoming drug resistance of malignant tumors. As key members of the β-1,4-GalTfamily, β-1,4-GalT-I and β-1,4-GalT-V markedly promote chemoresistance in leukemia by enhancing Hedgehog pathway activation and upregulating drug efflux proteins, while β-1,4-GalT-III appears to inhibit tumor progression by facilitating CD8+ T-cell-mediated immune responses. These findings highlight the therapeutic potential of selectively targeting GalT isoforms to overcome drug resistance and bolster antitumor immunity.

#### 5.5.2. Hedgehog-Focused Targeted Therapies

Targeting the Hedgehog (Hh) signaling pathway represents a promising strategy in cancer therapy. Among its components, Smoothened (Smo), a membrane protein with GPCR-like properties, offers an attractive candidate drug due to its structural flexibility, allowing incorporation in GPCR inhibitor design strategies, as highlighted in Table 1 [144].

Several Hh pathway inhibitors have advanced to clinical trials across a range of malignancies. For example, BMS-833923 (XL-139) is being evaluated in Phase I and II trials for chronic myeloid leukemia (CML) [145].

Vismodegib (GDC-0449) has reached Phase II trials in various tumor types, including metastatic colorectal cancer (mCRC) [145]. However, in a Phase II mCRC trial, patients were administered with Vismodegib (150 mg/day orally) or placebo, in combination with FOLFOX or FOLFIRI chemotherapy plus bevacizumab (5 mg/kg) every 2 weeks until disease progression or intolerable toxicity. Vismodegib did not exhibit superior efficacy compared to standard therapy, with median progression-free survival (PFS) of 7.5 months versus 5.8 months for placebo [146].

Itraconazole, traditionally an antifungal agent, has been repurposed for its inhibitory effectiveness on the Hh pathway, and is currently in Phase II trials as a treatment for basal cell carcinoma (BCC) [147]. In a Phase II trial, patients who had at least one basal cell carcinoma (BCC) tumor greater than 4 mm in diameter were enrolled onto two cohorts to receive oral itraconazole 200 mg twice per day for 1 month (cohort A) or 100 mg twice per day for an average of 2.3 months (cohort B). Itraconazole treatment yielded a 45% reduction in cell proliferation, a 65% decrease in Hh pathway activity, and a 24% reduction in tumor area. In this study of eight patients with multiple tumors, four showed partial responses, and four maintained a stable level of disease [148].

Taladegib (LY2940680) is engaged in Phase I and II evaluations for small-cell lung carcinoma, esophageal junction cancer, and idiopathic pulmonary fibrosis, indicating potential applications beyond oncology [149]. In Phase I trials, LY2940680 was administered orally once daily in continuous 28-day cycles, with treatment continued until either disease progression or unacceptable toxicity. Patients remained on therapy for variable periods ranging from several weeks to over 12 months, depending on individual tolerability and response. The maximum tolerable dose (MTD) was determined at 400 mg. Common adverse effects included dysgeusia, fatigue, muscle spasms, and nausea. Among patients, 30.9% achieved clinical benefit, 28.6% showed disease progression, and 26.2% experienced other responses. A Phase II study was recommended due to the small sample size [147].

Sonidegib (Erismodegib, LDE-225) is being tested in Phase I and II trials for medulloblastoma, basal cell carcinoma, and multiple myeloma [149]. In a Phase I/II trial in relapsed medulloblastoma (MB), patients with recurrent intracranial meningiomas were treated with oral sonidegib (LDE225) at a dose of 800 mg once daily in continuous 28-day cycles. Treatment was continued until disease progression or the development of unacceptable toxicity. Total inhibitions were observed in four patients (two pediatric and two adult), with one additional partial response. The recommended Phase II dose for pediatric patients was established at 680 mg/m^2^ [150].

Collectively, these trials underscore the therapeutic promise of Hh pathway inhibition across both solid and hematologic malignancies, supporting its potential as a versatile tool for precision oncology.

The Hh pathway is involved in vital processes including embryogenesis and somatic development. However, aberrant activation of the Hh pathway is associated also with carcinogenesis and CSC regulation. Emerging evidence suggests that inhibiting this pathway can be an effective strategy for cancer treatment. Inhibition of β-1,4-GalT-V or β-1,4-GalT-I in cancer cells (HL-60) has been shown to inhibit the Hh pathway, resulting in inhibition of Hh tumorigenesis. Future research is needed to elucidate the effect of the β-1,4-GalT family, especially focusing on β-1,4-GalT-V and β-1,4-GalT-I, which can play a vital role in activating such cancer-related pathways.

### 5.6. β-1,4-GalT-V Interaction with Snail-1

#### 5.6.1. Background

Snail-1 is one of the three zinc finger factors of the Snail transcription family [151]. This transcription factor is named for its ability to control gene activity by attaching to DNA. Snail-1 induces chemokine expression, promoting the recruitment of immunosuppressive cells into the tumor microenvironment. This results in epithelial–mesenchymal transition (EMT) in which epithelial cells lose polarity and adhesion. Snail-1 promotes EMT by downregulating the expression of target proteins that inhibit EMT (Figure 7). Among diverse target proteins, E-cadherin is the most significant as it plays a role in many cellular functions, e.g., cell–cell adhesion. Thus, they acquire a mesenchymal phenotype. This promotes a microenvironment to allow invasive activities of the cells. Furthermore, Snail-1, Snail-2, and Snail-3 have highly homologous structures and functions. Snail-2 is an EMT-inducing transcription factor but it also plays a role in driving neoplastic epithelial cells. Snail-3 is a DNA-binding transcription factor and is concurrently involved in copper ion binding [151]. However, Snail-1 plays a central role in EMT regulation and directly relates to cancer progression. The involvement of Snail-1 in tumor progression follows a positive feedback loop. PD-L1 accumulates on tumor cell surfaces, interacting with PIP1B, which activates the p38-MAPK pathway that inhibits degradation of Snail-1 by GSK3β [38]. Consequently, Snail-1 promotes tumor metastasis through EMT. It also influences the tumor microenvironment by inducing the attraction of immunosuppressive cells such as M2 macrophage, neutrophil, and T-cells [152].

#### 5.6.2. Relationship Between Snail-1 and β-1,4-GalT-V

β-1,4-GalT-V is upstream to a cascade of pathways activated by LacCer and stimulates diverse cellular phenotypes, including proliferation and cell migration via generation of superoxides. A key pathway activated by superoxides is the Ras signaling pathway. ERK is a primary effector involved in cell migration, proliferation, and differentiation, and it is downstream of this pathway. ERK phosphorylates TCF transcriptional factors that lead to the expression of immediate early genes. Subsequently, they activate downstream targets to promote cell migration by inducing EMT [136], with Snail-1 serving as the key regulator.

#### 5.6.3. Relationship Between Snail-1 and CRC

Colorectal cancer (CRC) is a human malignancy that Snail-1 affects significantly. Due to the proximity between intestinal tissues, the metastatic abilities of colon cancer cells are closely associated with EMT and its related activation factors. After having undergone EMT, 77% of CRC samples were shown to have Snail-1 immunoreactivity in activated fibroblasts and carcinoma cells [153]. In addition, Snail-1 demonstrated a direct correlation with E-cadherin downregulation.

EPHB3 is a guidance factor involved in suppressing tumors in CRC. From a sample of stage I–IV colon and rectum carcinomas, Snail-1 was shown to trigger the displacement of ASCL2 in the endogenous EPHB3 locus. This induces a reaction cascade leading to downregulation of EPHB3 [153]. CRC is known for its high variability in mutations that also occur more extensively compared to heart-related conditions. Specifically, because CRC typically remains asymptomatic until it is in late stages, a recent study suggested Snail-1 overexpression to be a potentially informative biomarker for CRC [137].

#### 5.6.4. Snail-1 Prospects

While Snail-1 can contribute to cell survival and healing, it can also be involved in potentiating tumor growth. For example, Snail-1-mediated EMT can contribute to muscle remodeling during development or in response to injury. However, in colorectal cancer screening, Snail-1 overexpression could serve as a useful marker for high cancer proliferation [154], and its multifaceted involvement is being used to develop cancer treatments as well. Snail-1 is emerging as a novel strategy in an increasing number of chemotherapies.

The synthetic small molecule compound CYD19, combined with conserved 174 arginine, also represents a potentially potent therapeutic for manipulating various Snail-1 effects. CYD19 was shown to interfere with Snail acetylation, promoting its degradation via the ubiquitin-proteasome pathway, and tumor growth was also shown to be reduced [151]. It is currently under trials with mice.

In a clinical trial using Polyethylene Glycol 3350 (Table 1) to target Snail-1 for patients at risk of colorectal cancer, the expression of Snail-1 was quantified with RT-PCR, although Snail-1’s role was found to be secondary to EGFR [153]. Nevertheless, blocking Snail-1 showed a decrease in EMT progression, fewer immunosuppressive cells infiltrating the tumor microenvironment, and increased treatment effectiveness. Further advancements in blocking Snail-1 expression are becoming an encouraging approach to alleviating cancer progression.

### 5.7. Is β-1,4-GalT-V Engaged with PD-L1 Glycosylation?

#### 5.7.1. Background

PD-L1 (Programmed Death Ligand -1) is expressed in various tumors, while PD-1 (Programmed cell Death protein 1) is primarily expressed in T-cells within tumor tissues [155]. The binding of PD-L1 to PD-1 creates a molecular barrier that inhibits the cytotoxic actions of immune cells. It is possible to overcome this inhibition by blocking antibodies or using recombinant proteins that target specific signaling pathways and reactivate immune responses. Monoclonal antibodies against PD-1 and PD-L1 have demonstrated significant therapeutic success, suggesting that immune checkpoint blockade therapy can be a potent antitumor treatment.

PD-L1 is a glycoprotein encoded by the *CD274* gene on chromosome 9p24.1. Its structure includes a signal peptide, extracellular IgV and IgC domains, a transmembrane domain, and a cytoplasmic domain. The extracellular region contains four glycosylation sites, N35, N192, N200, and N219, which undergo N-linked glycosylation, a post-translational modification of Asparagine residue (Figure 8) [156]. Recent studies have shown that the PD-L1 intracellular domain (ICD) functions as an RNA-binding protein. The ICD of PD-L1 plays an important role in immune regulation in cancer by suppressing T-cells through the signals to its extracellular domain. The ICD consists of three conserved sequence motifs termed “RMLDVEKC”, “DTSSK”, and “FEET”. The distal part of RMLDVEKC and the entire DTSSK motif are placed within the RNA polymerase-like motif. The RMLDVEKC motif is essential for cancer cells to resist the apoptotic capacities of type I and II interferons and acts as a molecular shield. DTSSK and lysine residues in the carboxy terminus are negative regulators of PD-L1 activity. This effectively helps cancer cells resist immune-mediated attack. Thus, along with PD-L1/PD-1 pathway, co-targeting the pathways (m-TOR) involving these motifs could be beneficial for designing combinational immune therapies with existing or designing new therapies [157]. The fully glycosylated PD-L1 has a molecular weight of approximately 50 kDa, while the non-glycosylated PD-L1 is ~30 kDa. β-1,4-GalT-V could interact with the PD-L1 protein by glycosylating one of the Asp residues through its glycosyl transferase activity (Figure 8).

The post-translational modifications of PD-L1, like glycosylation, ubiquitination, phosphorylation, and palmitoylation, affect upregulation of PD-L1 in the tumor microenvironment. PD-L1 glycosylation was shown to potentiate its interaction with PD-1, leading to the suppression of T-cell toxicity (in the breast cancer model). N-glycosylation is a common PTM that occurs on the NXS/T motif of proteins in ER, Golgi apparatus. Studies in breast cancer cells show that N-Glycosylation of PD-L1 has a significant regulatory effect on the ability to avoid immune surveillance. Thus, PD-L1 glycosylation is notable for its immunosuppressive function. PD-1 is highly N-glycosylated in T-cells at N49, N58, N74, and N116 sites, whereas PD-L1 is glycosylated at N35, N192, N200, and N219 on extracellular domains in ER. N-glycosylation plays a critical role in the activity and stability of these proteins [158]. Also, studies in cancer cells treated with tunicamycin suggest that PD-L1 is exclusively N-glycosylated at N35, N192, N200, and N219. It is instructive that glycosylation of PD-L1 was completely inhibited when the cells were treated with tunicamycin in a dose-dependent manner but not when treated with O-glycosidase, indicating that PD-L1 is N-glycosylated [19]. The non-glycosylated PD-L1 protein exhibits extreme instability with a half-life of ~4 h. Serine/threonine phosphorylation of non-glycosylated PD-L1 mediates its ubiquitination and subsequent degradation. β-1,4-GalT-V possibly plays a role in post-translational modifications (N-glycosylation) of PD-L1 protein, affecting its immune regulatory response through immune checkpoint inhibition with PD-1 in T-cells.

Our previous studies showed that β-1,4-GalT-V KO in HUVECs contributed to a ~70% decrease in β-1,4-GalT-V gene/protein ablation and significantly mitigated VEGF-induced PECAM-1 gene expression and angiogenesis. In cold tumors like colorectal cancer, cancer cells metastasize using the angiogenesis pathway. As PD-L1 also plays a role in regulating angiogenesis, it can contribute to tumor growth and metastasis when promoting angiogenesis. PD-L1 can interact with VEGFR2, a receptor involved in angiogenesis, leading to the activation of signaling pathways like FAK/AKT, which can promote angiogenesis and tumor progression. Combining PD-L1 inhibitors with antiangiogenic drugs can be a promising strategy for cancer treatment, potentially enhancing the effects of both therapies. This approach can target both tumor vasculature and its immune-suppressive environment [159].

#### 5.7.2. PD-1-PD-L1 Immune Checkpoint Pathway

Immune checkpoints are a type of signal that regulates the antigen binding to T-cell receptors during the process of the immune response. PD-L1 interacts with PD-1 and inhibits the expression of various TFs of T-cells, such as GATA3 and T-bet. PD-L1 expression plays a significant role in the differentiation of regulatory T-cells and affects the maintenance of suppressive roles. Tregs are important inhibitors of tumor-specific immune responses in the TME, thus facilitating PD-1 and PD-L1 immune checkpoint pathways and helping to generate the Tregs. Tregs can protect the TME by forming a layer for immune evasion of tumors. Thus, blocking PD-1 and PD-L1 can activate antitumor activity through activation of T-cells and inhibition of Treg cells, and it can also promote CD4+T cell differentiation into FOX P3+ Tregs by enhancing suppression of the immune system, resulting in immune tolerance in cancer patients [160].

#### 5.7.3. The Regulatory Mechanism of PD-L1 Expression

Speculation suggests that PD-L1 plays a significant role in suppressing the adaptive immune system during events such as pregnancy, tissue allografts, and autoimmune disease. PD-L1 is an essential immune checkpoint protein that binds to PD1 (programmed death 1) on T-cells. T-cells play a key role in killing cancer cells, while cancer cells exhibit immune escape by the expression of PD-L1, which binds to PD-1 and gives a signal to T-cells to ignore cancer cells regarding apoptosis. PD-L1 acts as an off-switch for the immune system. Findings indicate that the inhibition of MYC gene expression in mouse/human tumor cells can reduce the expression of PD-L1 at both the gene and protein levels. PD-L1 expression is also regulated by the MAPK and PI3K/Akt signaling pathways. The effect of the PD-1 and PD-L1 interaction is important for co-inhibition during T-cell initiation of an immune response [161,162,163]. Upon T-cell activation, the resulting cytokines TNF-α and IL-4 also upregulate PD-1 ligands, establishing a feedback loop that attenuates immune responses and limits the extent of immune-mediated tissue damage.

#### 5.7.4. Immunotherapy by Targeting PD-1/PD-L1 Immune Checkpoint Pathway

Monoclonal antibodies such as Nivolumab, Pembrolizumab, Cemiplimab (PD-1 inhibitors), Atezolizumab, Avelumab, and Durvalumab (PD-L1 inhibitors) are commercially available for antibody therapy targeting the PD-1/PD-L1 pathway. This blockade inhibits interactions between PD-1 (found on T-cells) and PD-L1 (found on tumor cells), which can prevent tumor cells from escaping the immune system, further assisting T-cells in destroying them. These antibodies can act by targeting specific sites of either PD-1 or PD-L1 and preventing their binding. A challenging immunotherapy dilemma surrounds tumor specificity within populations at various stages of these diseases. However, the results from various clinical trials using combination therapies are more effective in the early stages of disease with apparently reduced benefits at advanced-stages of cancer [164].

Studies of patients from the Indian sub-continent with advanced NSCLC (non-small-cell lung cancer) show evidence of PD-L1 expression, associated with tobacco use exhibiting aggressive tumor characteristics. To advance our understanding of the scope and impact of tobacco-associated cancers, focused studies are needed specifically to elucidate the role of PD-L1 as an expedient prognostic marker in advanced NSCLC [165].

A study of 1156 NSCLC specimens, including 827 sequentially recently resected specimens and 293 biopsy specimens, showed that high PD-L1 expression was observed in 9.7% of 827 NSCLC patients, including 6.5% with adenocarcinoma (ADC, n = 690) and 27.4% with squamous cell carcinoma (SqCC, n = 117). These results showed higher expression rates than those in archived samples (>5 years old, n = 329) that were previously reported by the same group (4.9%, 0.5%, and 13.9% in NSCLC, ADC, and SqCC, respectively). The widespread presence of PD-L1 expression was lower in surgical resection samples than in small biopsy samples. PD-L1 high expression in the lung biopsy was less likely to be present in primary cancer than in metastases and was also associated with a high level of stromal TILs (*p* = 0.029) and PD-L1-positive immune cells (IC) (*p* < 0.001). Both PD-L1 high and low expressions were more frequent in the EGFR-wild type than in the mutant type (*p* < 0.001) [166].

Thus, PD-L1, is a formidable candidate for developing novel immunotherapies alone or combined with pre-existing drugs to mitigate solid-tumor-bearing cancer (Table 1).

### 5.8. β-1,4-GalT-V Interaction with β-Catenin

#### Background

β-1,4-GalT-V is known to regulate the stemness (the potential to give rise to other cells) of breast cancer stem cells (BCSCs) and is upregulated by the Frizzled-1 receptor while persistently activating Wnt/β-catenin signaling. The evidence shows that β-catenin is an important precursor to cancer pathways. In colon cancer, β-catenin inhibits NF-kβ DNA-binding, transactivation activity, and target gene expression. In hepatocellular carcinoma, β-catenin participates in various stages of hepato-carcinogenesis. In gastric cancer, increased levels of β-catenin mRNA have been reported, as well as mutational alterations of antigen-presenting cells (APCs) and the β-catenin gene [5]. In the nucleus, T-cell factor/lymphoid enhancer factor (TCF/LEF) TFs recruit β-catenin to target Wnt genes, causing transcription activation and transcriptional coactivators [19]. In this process, Tcf factors function as tumor inducers and aberrantly target genes due to rising β-catenin levels, as expressed in many types of cancer [167]. Abnormal β-catenin levels can be caused by stabilized mutations within β-catenin or by truncating mutations in the adenomatous polyposis coli (APC) tumor suppressor gene. Although this study focused on β-1,4-GalT-V and β-catenin, these results have stimulated interest regarding other cancers, like colorectal cancer [151]. According to an article from the AHA, during cellular migration, β-catenin does not bind to cadherins, as previously believed [167]. Instead, it leads to a loss in tissue integrity by translocating to the nucleus. It was also reported that increased level of β-catenins can lead to increased Runx2 expression, which subsequently enhances ossification, contributing to atherosclerosis [168].

### 5.9. β-1,4-GalT-V Interaction with P62 and Autophagy

P62 is an autophagy receptor and a multifaceted cellular protein involved in various signal transduction pathways, like the Keap1-Nrf2 pathway. When cellular p62 levels are altered, the number and location of ubiquitinated proteins are impacted, affecting cell survival, specifically causing proteasomal degradation of ubiquitinated proteins [165]. Studies show that mice with cigarette-smoke-induced emphysema and lung tissue from COPD patients have significantly increased levels of LacCer with increasing severity of emphysema. This was accompanied by an increase in the protein expression of p62, a biomarker for defective autophagy. Conversely, treatment with D-PDMP, a GlcCer synthase/β-1,4-GalT-V inhibitor, reduces the number of p62-positive cells triggered by cigarette smoke extract (CSE) and Pseudomonas aeruginosa lipopolysaccharide (Pa-LPS) during acute sub-chronic exposure, compared to the EF control.

To model the role of tumor suppression in autophagy, a hepatic cancer framework was applied, including overexpressed p62 in mice heterozygous for the autophagy regulator Beclin 1 [167]. In this study, autophagy was partially inhibited, and spontaneous tumors developed. TNF-α-induced canonical signaling through the proinflammatory transcription factor NF-κB was suppressed due to p62 accumulation. This suppression of TNF-α-induced NF-κB signaling was associated with P62 accumulation, showing an inverse relationship between p62 levels and inflammatory activity in tumors of *Beclin 1^+/−^* mice. Additional studies show that blocking GlcCer and LacCer synthesis using D-PDMP or Eliglustat alters autophagy in osteoclasts, thus improving myeloma bone disease [169]. Further autophagic degradation of TNF receptor associated factor (TRAF3), a key step in osteoclast differentiation, was inhibited by eliglustat, evidenced by TRAF3 action in restoring osteoclast formation in bone marrow cells in myeloma [170].

**Table 1 ijms-26-08088-t001:** Therapeutic approaches targeting the β-1,4-GalT-V Interactome.

Drug	Phase	Cancer	Target Pathway
Vantictumab [114]	Phase 1	Breast cancer	Frizzled
Ipafricept [116]	Phase 1	Ovarian cancer	
BMS-833923 (XL-139) [149]	Phase 1 and 2	Chronic myeloid leukemia	Hedgehog
Vismodegib (GDC-0449) [145]	Phase 2	Multiple tumors including colorectal cancer	
	Phase 2	mCRC	
	Phase 1/2	First line mCRC	
Itraconazole [147]	Phase 2	Basal cell carcinoma	
Taladegib (LY2940680) [149]	Phase 1 and 2	Small cell lung carcinoma	
	Phase 1 and 2	Esophageal junction cancer	
	Phase 1	Idiopathic pulmonary fibrosis	
Sonidegib (Erismodegib, LDE-225, NVP-LDE-225) [149]	Phase 1 and 2	Medulloblastoma	
	Phase 2	Basal cell carcinoma	
	Phase 2	Multiple myeloma	
WNT974 [101,102]	Phase 1, 1b, and 2	Melanoma, breast cancer, CRC, and pancreatic adenocarcinoma	Wnt-1 pathway
Polyethylene Glycol 3350 [171]	Phase 1 and 2	Colorectal cancer	Snail-1
Nivolumab/Ipilimumab [164]	Phase 1 and 2	NSCLC	
	Phase 1 and 2	Advanced solid tumors	PD-1/PD-L1 pathway
Pembrolizumab [164]	Phase 1	Squamous NSCLC	
	Phase 1 and 2	NSCLC	
	Phase 1	Advanced/metastatic non-squamous NSCLC	
Atezolizumab [164]	Phase 3	Metastatic NSCLC	
	Phase 2	NSCLC stage 3 and 4	
Sintilimab [164]	Phase 2	Advanced NSCLC	
Avelumab [164]	Phase 1 and 2	Metastatic NSCLC	
Toripalimab [164]	Phase 2	NSCLC	

These findings provide a compelling molecular explanation for the persistent inflammatory state observed in smokers, independent of any assessment of direct DNA damage. Moreover, they establish β-1,4-GalT-V as a promising therapeutic target for reversing cigarette-smoke-induced immune defects, particularly relevant in chronic inflammatory diseases and cancers such as colorectal cancer, where β-1,4-GalT-V is already implicated in tumor progression and altered glycosphingolipid signaling.

This body of work bridges environmental exposure, immune regulation, and glycosylation biology, offering novel avenues for intervention in both cancer and smoke-induced chronic inflammation.

### 5.10. Perspectives

This article presents an overview of the interaction of β-1,4-GalT-V with various key signaling molecules, playing a pivotal role in activating life-threatening disease pathways. This research could pave the way for potential biomarker exploration and novel treatments, such as the early diagnosis of colorectal cancer and more effective therapeutics. Future research should investigate whether β-1,4-GalT-V could be an ideal marker for such disease phenotypes, and whether inhibiting β-1,4-GalT-V upstream may lead to side effects in such disease models. The role of this enzyme in drug resistance must be clarified, as lactosylceramide acts as a secondary messenger for many pathways involved in stem cell differentiation, angiogenesis, cell growth, and cell proliferation. Developing a therapeutic approach to inhibit this enzyme is of paramount importance. Furthermore, targeting β-1,4-GalT-V’s interactions has become critical as an outlook in more effectively combatting multidrug resistance in novel cancer treatments. Based on the information provided, it is possible that β-1,4-GalT-V has an important role in breast and prostate cancer as well.

## 6. Conclusions

Over the past several decades, Dr. Chatterjee’s lab and many others have been elucidating β-1,4-GalT-V’s role and function linked to physiological and pathological processes. Our lab’s contributions are as follows. We demonstrated that various physiologically relevant molecules and other health-affecting factors such as Western diet, cigarette smoke, and stress can converge upon β-1,4-GalT-V to generate LacCer. LacCer is a “bonafide” bioactive signaling molecule that can activate the “oxidative stress pathway” leading to angiogenesis, proliferation, migration, phagocytosis, and apoptosis—all the critical cellular phenotypes in health and disease—and the LacCer-induced “inflammatory pathway” affects skin diseases, inflammatory bowel disease, and COPD, among others.

Sp1 is a nuclear factor and a transcriptional regulator of β-1,4-GalT-V.β-1,4-GalT-V regulates VEGF-independent angiogenesis by generating LacCer and galactosylation of Notch-1, regulating production of glioma-like stem cell differentiation into endothelial cells and promoting tumorigenesis. These observations suggest a VEGF-independent pathway contributing to angiogenesis, a phenotype critical in tumor metastasis and atherosclerotic plaque growth, and plaque stability via inducing mature neo-vessels and monocyte/neutrophil infiltration.Oxidative stress increases LDL oxidation to form oxidized LDL that enters cells via an LDL-receptor-independent pathway/scavenger pathway. Oxidized LDL phosphorylates serine, threonine, and tryptophan in β-1,4-GalT-V, thus generating LacCer, causing downstream activation of critical phenotypes in cultured vascular cells and blood vessels in ApoE-/- mice fed a Western diet. In turn, this leads to atherosclerosis, cardiac hypertrophy, and atherosclerotic plaque development. Conversely, feeding D-PDMP or a biopolymer-encapsulated D-PDMP (inhibitor of GlcCer synthase and LacCer synthase) reverses atherosclerosis and cardiac hypertrophy, and improves vascular and cardiac functions.The Snail-1 protein plays a vital role in inducing EMT and tumor progression by inhibiting cell-adhesion proteins, promoting immunosuppressive tumor microenvironment, and influencing PD-L1 accumulation. This inhibits the p38-MAPK pathway that degrades Snail-1. The Ras signaling pathway is activated by superoxides, a downstream upregulation of β-1,4-GalT-V. This pathway involves ERK, which is involved in the promotion of EMT, a process regulated by Snail-1.Cigarette smoke increases LacCer accumulation in bronchial epithelial cells and macrophages as well as in the lungs in patients with COPD. This is accompanied by the increased expression of defective-autophagy marker p62. Conversely, treatment with D-PDMP reversed the pathology in mice subject to cigarette smoke. Additionally, blocking GlcCer and LacCer synthesis using D-PDMP or Eliglustat alters autophagy in osteoclasts, improving myeloma bone disease.β-1,4-GalT-V protects FZD_1_ from degradation, possibly via N-linked glycosylation, activating the Wnt/b-Catenin pathway. PD-L1 also activates the β-catenin signaling pathway, causing β-catenin to bind to TCF4/β-catenin binding sites on the PD-L1 promoter. Thus, PD-L1 and β-catenin form a positive feedback loop in regulating the expression of target genes such as stem cell markers and calcification. β-1,4-GalT-V also promotes cell proliferation via the Hedgehog pathway.β-1,4-GalT-V’s influence on these pathways posits it as a promising therapeutic agent as well as a diagnostic marker in colorectal cancer and many other cancers. Our future research aims are to define mechanisms fundamental to β-1,4-GalT-V’s regulatory effects and to further investigate therapeutics that target β-1,4-GalT-V in treating cancer, cardiovascular diseases, and inflammatory diseases.

## Figures and Tables

**Figure 1 ijms-26-08088-f001:**
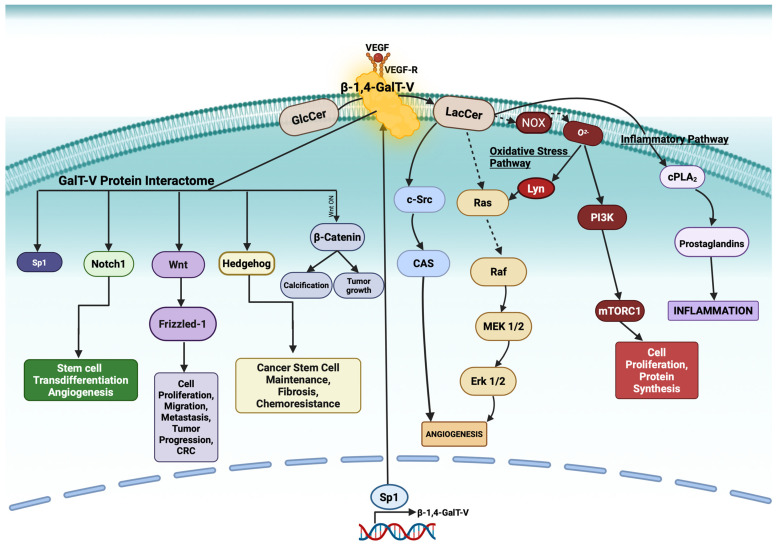
A schematic roadmap illustrating how β-1,4-galactosyltransferase-V (β-1,4-GalT-V) regulates cellular signaling through the glycosylation of both lipids and proteins. β-1,4-GalT-V catalyzes the transfer of galactose to glucosylceramide (GlcCer), forming lactosylceramide (LacCer), a bioactive glycosphingolipid. Activation of the VEGF receptor (VEGFR) enhances β-1,4-GalT-V activity and LacCer production. LacCer in turn activates NADPH oxidase (NOX2), generating reactive oxygen species (ROS) and initiating oxidative stress signaling via Lyn kinase. This cascade stimulates PI3K and mTORC1/2 pathways, promoting cell proliferation, protein synthesis, and survival. LacCer also activates cytosolic phospholipase A2 (cPLA2), leading to prostaglandin-mediated inflammation. Additionally, LacCer activates the Raf/MEK/ERK pathway and focal adhesion signaling through c-Src and CAS, contributing to angiogenesis. Protein glycosylation by β-1,4-GalT-V may indirectly influence key regulatory proteins, including Sp1 (a transcription factor), Notch1, and β-catenin, affecting transcriptional control, Wnt signaling, and stem cell fate. Created with BioRender. Adapted with permission from [4,5].

**Figure 2 ijms-26-08088-f002:**
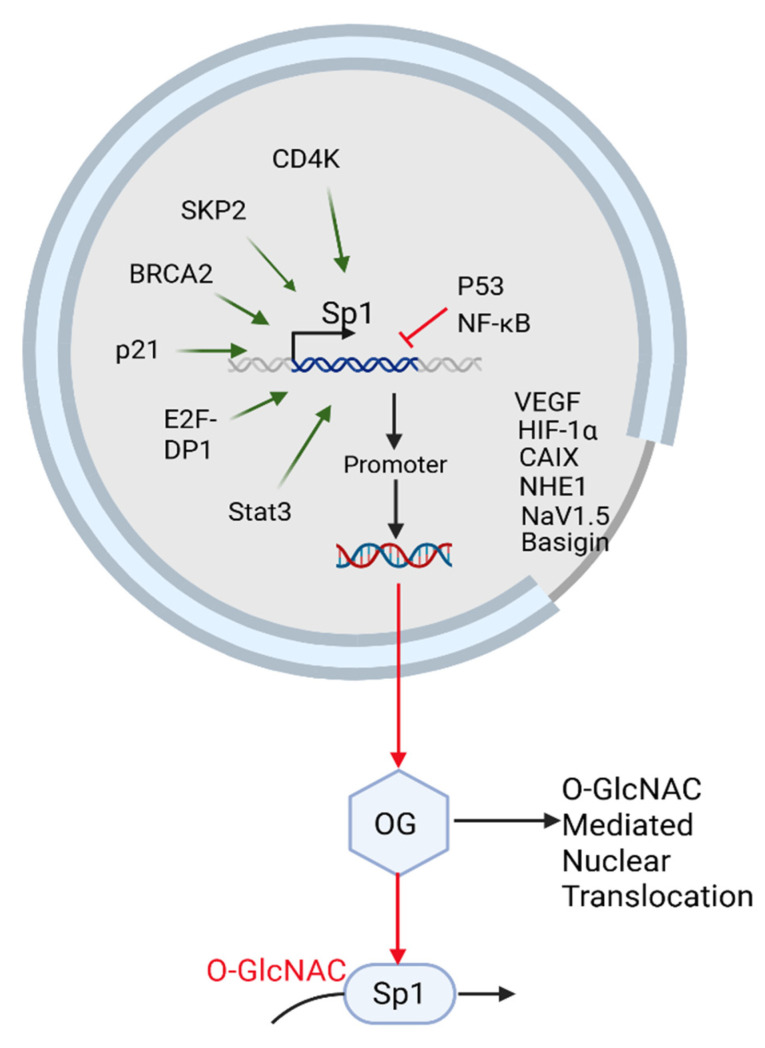
**Schematic representation of Sp1 transcription factor regulation and its downstream gene targets in the nucleus.** Sp1 activity is modulated by upstream activators, including CDK4, SKP2, BRCA2, p21, E2F-DP1, and Stat3 (green arrows), while it is repressed by p53 and NF-κB (red blunt pointers). Sp1 binds to promoter regions to activate transcription of key genes involved in tumor progression, pH regulation, and immune modulation, including VEGF, HIF-1α, CAIX, NHE1, NaV1.5, and Basigin. Additionally, SP1 undergoes O-GlcNAcylation via the hexosamine biosynthetic pathway (HBP), a modification that facilitates its nuclear translocation and enhances transcriptional activity (red arrows) (Created with BioRender).

**Figure 3 ijms-26-08088-f003:**
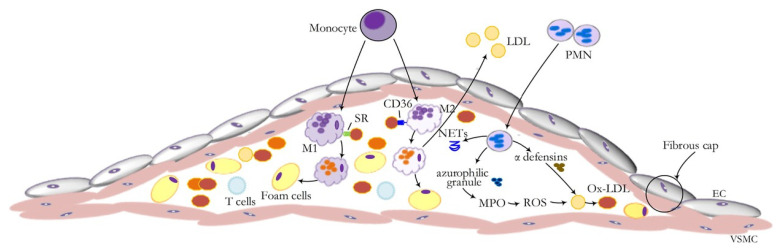
Inflammatory activation of vascular endothelium during atherosclerosis. Ox-LDL and LPSs (lipopolysaccharides) promote immune cells to release cytokines like TNF-α. These cytokines activate endothelial cells and increase the activity of β-1,4-GalT-V and promote immune cells to the inflammation site. Sp1, although not labeled, controls expression of adhesion molecules (VCAM-1, ICAM-1), inflammatory mediators, and SRs (scavenger receptors) involved in ox-LDL uptake. This figure shows the downstream transcriptional pathway that Sp1 moves through within the vascular endothelium. Specifically, promoting plaque formation, SRs on the surface of M1-type macrophages recognize and engulf ox-LDL to form foam cells. CD36 receptors help macrophages take in ox-LDL, which can lead to the formation of foam cells, contributing to atherosclerotic plaques in blood vessels. CD36 can also help fatty acid oxidation in anti-inflammatory M2-type macrophages, supporting tissue repair and lipid output. Simultaneously, α defensins and azurophilic granules and LacCer released by neutrophils further oxidize LDL to produce more ox-LDL. Ox-LDL overproduction causes lipid accumulation and, thus, forms more foam cells. EC: endothelial cell; VSMC: vascular smooth muscle cell; M1: M1-type macrophages; M2: M2-type macrophages; LDL: low-density lipoprotein; ox-LDL: oxidized low-density lipoprotein; MPO: myeloperoxidase; ROS: reactive oxygen species; PMNs: polymorphonuclear neutrophils; NETs: neutrophil extracellular traps. Adapted with permission from Ref. [56].

**Figure 4 ijms-26-08088-f004:**
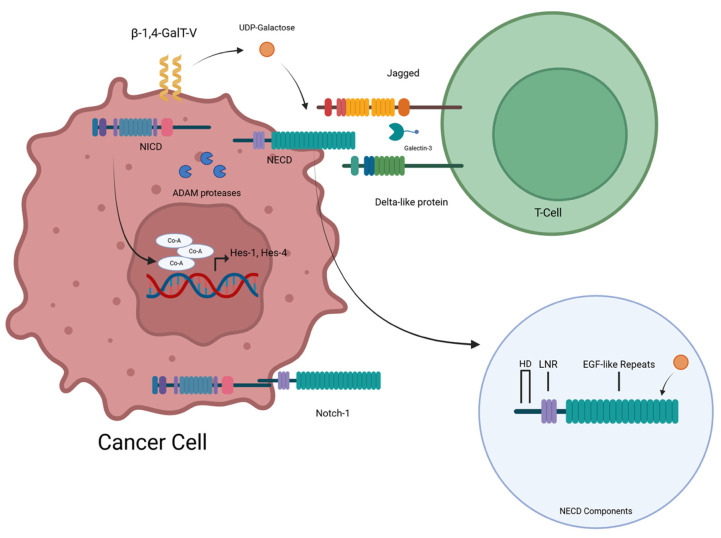
β-1,4-GalT-V–mediated glycosylation of Notch-1 in endothelial cells. Schematic representation of β-1,4-Galactosyltransferase V (β-1,4-GalT-V)-mediated glycosylation of Notch-1 and its role in T-cell–cancer cell interaction via the Notch signaling pathway. In the cancer cell, Notch-1 is synthesized and inserted into the plasma membrane. β-1,4-GalT-V utilizes UDP-galactose to glycosylate the extracellular domain of Notch-1, particularly at the epidermal growth factor (EGF)-like repeats, as shown in the magnified NECD components inset (highlighting HD: heterodimerization domain; LNR: Lin12/Notch repeats; and EGF-like repeats). This glycosylation promotes interaction of Notch-1 with T-cell–derived ligands such as Jagged and Delta-like proteins, which are further stabilized by Galectin-3 binding. Ligand binding triggers cleavage of Notch-1 by ADAM proteases, releasing the Notch intracellular domain (NICD), which translocates to the nucleus. NICD interacts with transcriptional co-activators (Co-A) to induce the expression of Notch target genes, including Hes-1 and Hes-4. The figure summarizes key molecular components involved in the Notch signaling axis between T-cells and cancer cells, including β-1,4-GalT-V, UDP-galactose, Galectin-3, Notch ligands (Jagged, Delta-like), and the modular domains of the Notch extracellular domain (NECD). Created with permission from Hideyuki Takeuchi and with Biorender [75].

**Figure 5 ijms-26-08088-f005:**
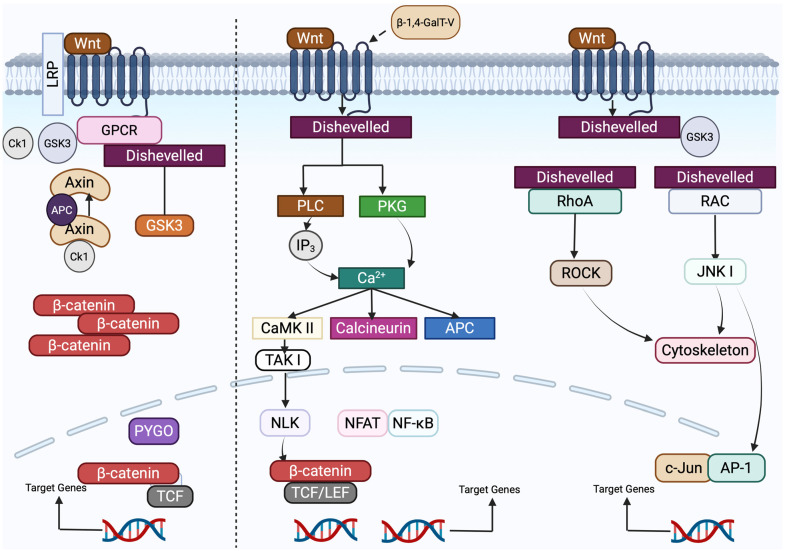
**Schematic Illustrating canonical and non-canonical Wnt pathways depicting tumor progression.** In the absence of Wnt ligands, cytoplasmic β-catenin is phosphorylated by a destruction complex comprising GSK3, casein kinase I (CK1), Axin, and APC. Phosphorylated β-catenin is subsequently targeted for proteasomal degradation. Upon Wnt ligand binding to the Frizzled-Lipoprotein Receptor-Related Protein (LRP) receptor complex, Dishevelled (Dvl) is recruited, leading to inhibition of the β-catenin destruction complex. This allows cytoplasmic β-catenin to accumulate and translocate into the nucleus, where it partners with TCF/LEF to drive transcription of Wnt target genes (canonical Wnt/β-catenin pathway). Additionally, Wnt can activate non-canonical pathways: the Wnt/Ca^2+^ pathway, which involves PLC, PKG, and Ca^2+^-dependent effectors such as CaMKII, calcineurin, and NFAT; and the planar cell polarity (PCP) pathway, where Dvl activates small GTPases (RhoA and Rac), leading to cytoskeletal remodeling through ROCK and JNK. Notably, β-1,4-Galactosyltransferase-V (β-1,4-GalT-V) may influence Wnt signaling by modulating Wnt ligand interaction. Created with BioRender.com. Taken with Permission from Ref. [86].

**Figure 6 ijms-26-08088-f006:**
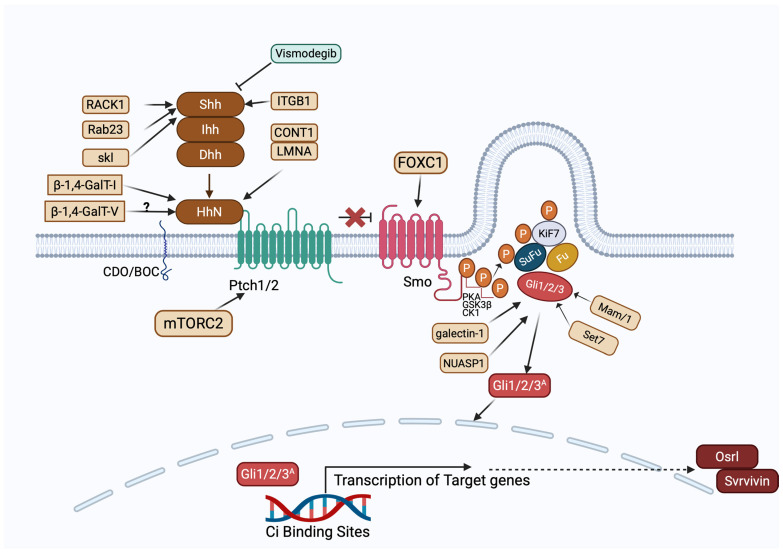
The binding of Hedgehog ligands (Sonic Hedgehog (Shh), Indian Hedgehog (Ihh), or Desert Hedgehog (Dhh)) to the Patched receptors (Ptch1/2) relieves their suppression of Smoothened (Smo), allowing phosphorylation of multiple Ser/Thr residues at the C-terminus of Smo. This activates Smo and triggers a cytoplasmic signaling cascade involving Kinesin Family Member 7 (KIF7), Suppressor of Fused (Sufu), and other factors, resulting in the nuclear translocation of Gli transcription factors (Gli1/2/3) and the transcription of downstream target genes such as *Osr1* and *Survivin*. Regulators such as β-1,4-Galactosyltransferase I (β-1,4-GalT-I) promote pathway activation through galactose transfer, and β-1,4-GalT-V is hypothesized to play a similar role. Other modulators include mTOR complex 2 (mTORC2), Forkhead box C1 (FOXC1), and membrane-associated proteins such as integrin β1 (ITGB1), Lamin A/C (LMNA), and CONT1. The small-molecule Smo inhibitor vismodegib blocks pathway activation and downstream effects (created with BioRender).

**Figure 7 ijms-26-08088-f007:**
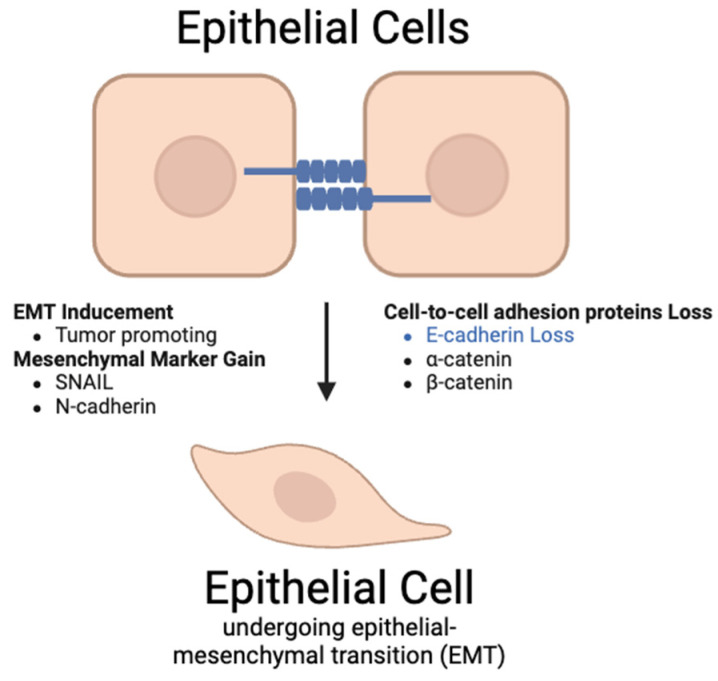
This figure depicts epithelial-to-mesenchymal transition (EMT), the process by which epithelial cells (**top**) transform into mesenchymal cells (**bottom**). The bullet points underline the key characteristics of EMT. SNAIL and N-cadherin are upregulated, and E-cadherin, ɑ-catenin, and ꞵ-catenin are lost. The overall process promotes tumor growth. Created with Biorender.

**Figure 8 ijms-26-08088-f008:**
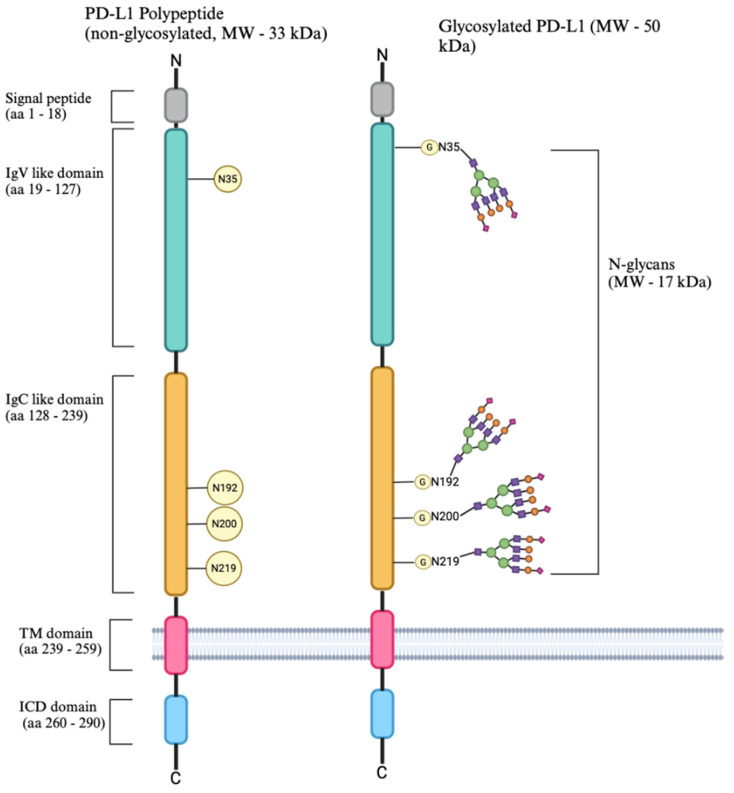
**The domain structure and glycosylation sites on PD-L1.** PD-L1 is a cell membrane protein with four glycosylation sites (G: yellow circles) of asparagine residues (N35, N192, N200, and N219) spanning the IgV-like and IgC-like domains of PD-L1. The numbers represent amino acid residues. The estimated molecular weight of PD-L1 polypeptide is about 33 kDa in a non-glycosylated form (**left**). Glycosylated PD-L1 consists of about 17 kDa of N-glycan moieties in a range of bands at about 50 kDa on Western blots (**right**). MW: molecular weight; IgV: immunoglobulin variable; IgC: immunoglobulin constant; TM: transmembrane; ICD: intracellular domain. Created with BioRender. Adapted with permission from [156].

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
