# Peer review of "The β-1,4 GalT-V Interactome—Potential Therapeutic Targets and a Network of Pathways Driving Cancer and Cardiovascular and Inflammatory Diseases"

_ijms, 2025, doi:10.3390/ijms26168088_

Round 1
Reviewer 1 Report
Comments and Suggestions for Authors
This manuscript provides a review of a highly relevant and cutting-edge topic. However, its structure and approach exhibit several notable deficiencies. It is imperative to condense several sections, as their excessive length detracts from the clarity and focus of the manuscript’s main objectives. Moreover, numerous paragraphs lack logical cohesion, while others unnecessarily separate related concepts into distinct paragraphs. Additionally, several paragraphs are devoid of appropriate citations, which is especially concerning in a review article and indicative of a lack of scholarly rigor. Furthermore, the figures are incorrectly numbered, reflecting a significant oversight, and the figure legends are insufficiently detailed. The following observations are outlined below:
- The title does not include a highly important topic addressed in the manuscript, namely the potential therapeutic targets.
- Line 11: I consider that it is more common to refer to this enzyme as β-1,4-Galactosyltransferase V (β-1,4-GalT-V); however, I leave this to the authors’ consideration. Additionally, it should be denoted with the Greek letter beta rather than the letter “B.”
- Line 14: It also produces and glycosylates high-branched N-glycans in the Golgi apparatus.
- Lines 16-30: This entire section can be omitted and removed, as it is a summary and therefore should remain concise, given that all of this content will be expanded upon in the main body of the review.
- Lines 32-37: The same observation as above applies; this section may be removed, as it is part of the summary.
- Figure 1: The explanation provided in the figure legend should be condensed and must correspond precisely to the content depicted in the illustration. For example, there is no mention of the VEGF receptor or how it activates the enzyme responsible for producing LacCer, nor is it explained that the enzyme catalyzes the transfer of galactose to glucosylceramide (GlcCer) to form lactosylceramide (LacCer). Furthermore, the activation of pathways such as MAPKs, PI3K/mTOR, among others, is not defined. Additionally, the manuscript does not clarify the meaning of the abbreviations used or specify whether they refer to kinases, transcription factors, or other entities.
- Lines 69-71: The wording of this statement should be revised, as it is not entirely accurate; GalT-V is not a direct synonym for LacCer synthase, although it may exhibit redundant activity.
- Lines 111-113: It is necessary to describe this statement in greater detail, as it lacks an explicit explanation in the figure. In the illustration, the enzyme is depicted in the membrane, but it is not specified whether this refers to the plasma membrane or the Golgi apparatus. Furthermore, the manuscript does not clarify how many transferases exist (five, six, etc.) or whether all of them produce LacCer.
- Line 135: It is necessary to specify which of these NOX isoforms is being referred to: NOX1, NOX3, NOX4, NOX5, DUOX1, or DUOX2.
- Lines 237-246: This paragraph lacks citations and presents ideas that are repetitive of those in the preceding paragraphs (lines 188–196).
- Lines 247-254: This paragraph lacks citations, particularly in the first two lines.
- Figure 4: The numbering is not consecutive, as the last figure was Figure 1. Such basic errors significantly undermine the quality of the manuscript, as they indicate that the authors did not take sufficient care to review their work prior to submission and appear to expect the reviewers to perform tasks that should have been completed beforehand. Furthermore, the manuscript presents a large number of factors, proteins, and other elements for which only abbreviations are provided, without indicating their full names or definitions.
- Line 272: I suggest rewriting this statement, as it cites only a single study conducted in one cell line yet generalizes the findings by referring to cancer cells in the plural. Alternatively, additional references should be included to support this claim.
- Line 274: The period following the word “signaling” should be removed.
- Lines 276-298: I suggest that all of these paragraphs be moved to the beginning, as they provide the context for the function of SP1. The preceding section is a specific example and, therefore, should be presented afterward.
- Line 325: As Latin is a dead language, de novo should be written in italics.
- Figure 3: The figure numbering should be reviewed and corrected as appropriate. Additionally, the meaning of the abbreviations is not provided. The manuscript also fails to explain what induces endothelial activation. Moreover, SP1 is not indicated in the figure in relation to the description provided.
- Lines 364-366: It is important to include references in this paragraph.
- Line 421: This paragraph should be placed at the end, as it addresses the potential therapeutic applications.
- Line 456: It would be highly appropriate to specify what each abbreviation refers to, as a matter of formality.
- Line 461: Figure 2 is not included in the manuscript.
- Line 474: It would be important to indicate it by its abbreviation, as in human monocyte chemoattractant protein 1 (MCP-1).
- Figure 4: A brief description of the figure should be included.
- Lines 494-495: It is important to indicate the meaning of the abbreviations.
- Line 498: To what does this indication refer? Is it part of the figure legend? It is unclear.
- Line 515: As in situ is a Latin term, it should be italicized.
- Line 519: It is not necessary to reference Figure 4 twice if the subsequent text also pertains to Figure 4.
- Line 522: It is necessary to specify what E1AF refers to.
- Line 552: If it is still unknown whether Notch1 participates independently of VEGF, then what is the rationale for providing several examples of VEGF’s involvement with GalT-V? This is unclear.
- Lines 568-570: This paragraph should be combined with the preceding one, as they express the same idea; grammatically, there is no justification for separating them.
- Lines 572-584: There are no citations throughout this entire paragraph; may I ask why?
- Lines 601-603: Is citation 2 in the first line of this paragraph the same one referenced elsewhere? Because there is no citation at the end of the paragraph, despite the statement being of significant importance.
- Line 618: It is important to specify the meaning of the abbreviation Treg, as was done for TGF-beta.
- Line 626: The citation is missing in this paragraph; is it the same as citation 84 referenced at the beginning of the paragraph? If so, this should be clearly indicated. Failing to specify citations reflects a significant lack of attention to detail.
- Figure 5: Please review the figure numbering, as the consecutive sequence is incorrect. Several proteins and enzymes depicted in the figure, such as calcineurin, c-Jun, Dishevelled, AP-1, among others, are not described in the figure legend.
- Line 649: I suggest removing this subtitle, as it is unnecessary; it is understood that a background of the pathway will be provided.
- Lines 650-654: The paragraph lacks citations.
- Line 660: The figure numbering does NOT correspond correctly to the sequence!
- Lines 700-706: Citations are missing! It is highly concerning that references are not provided throughout the manuscript in a review article, as this reflects significant negligence and lack of diligence.
- Lines 735-736: The statement lacks citations!
- Line 752: I suggest removing this subtitle, as it is unnecessary; it is understood that a background of the pathway will be provided.
- Line 769: Please review the figure numbering.
- Lines 785-788: Please review the figure numbering. Additionally, this paragraph conveys the same idea as the preceding one and, therefore, should not be separated but included together with the previous paragraph.
- Lines 794-799: The same recommendation as above applies in all respects.
- Lines 818-824: References are missing in this paragraph; these are important assertions.
- Lines 840-841: This paragraph is isolated and should be combined with the following one, as they convey the same contextual idea.
- Lines 842-843: It is important to specify the duration for which the drug was administered in this study.
- Figure 6: Please review the figure numbering and ensure it is consistent with that of the other figures. Several abbreviations present in the figure remain undefined, and a concise and appropriate description of the figure is lacking.
- Line 769: Please review the figure numbering.
- Lines 969-971: It would be advisable for the authors to provide a more detailed description of the study, as the current explanation is quite vague.
- Lines 994-995: This paragraph is part of the same idea as the following paragraph.
- Line 1036: There is a double period; please review.
- Figure 7: Please review the figure numbering and provide a more accurate description of the figure in accordance with its content.
- Line 1074: A citation is missing for the statement in the first line.
- Figure 9: Please review the figure numbering.
- Line 1086: Why was this topic not addressed when discussing the Wnt pathway? It seems out of context and repetitive, as several points mentioned here were already covered earlier during the discussion of the Wnt pathway.
- Lines 1210-1216: No references are provided in this paragraph. May I ask why?
- Lines 1225-1229: The assertion regarding the relationship between P62, autophagy, and the persistent inflammatory state in chronic smokers is unclear.
- Line 1247: Why are only prostate and breast cancers mentioned? Would these findings not be applicable to other types of cancer? The authors place significant emphasis on colorectal cancer.
- Line 1249: In the conclusion, the authors do not mention other pathways analyzed, such as autophagy, Snail1, Hedgehog, LDL, and atherosclerosis. The authors should strive to condense all the information, as they address a large number of topics, yet not all are included in the conclusion.
- Line 1284: Table 1 is not referenced in the text.
Author Response
- The title does not include a highly important topic addressed in the manuscript, namely the potential therapeutic targets.
Agreed, we have changed the title accordingly.
- Line 11: I consider that it is more common to refer to this enzyme as β-1,4-Galactosyltransferase V (β-1,4-GalT-V); however, I leave this to the authors’ consideration. Additionally, it should be denoted with the Greek letter beta rather than the letter “B.”
Agreed, we appreciate this reviewer’s comment. Hereafter, we shall refer this enzyme as β-1,4-Galactosyltransferase V (β-1,4-GalT-V).
- Line 14: It also produces and glycosylates high-branched N-glycans in the Golgi apparatus.
Agreed, we have fixed the mistake.
- Lines 16-30: This entire section can be omitted and removed, as it is a summary and therefore should remain concise, given that all of this content will be expanded upon in the main body of the review.
Done.
- Lines 32-37: The same observation as above applies; this section may be removed, as it is part of the summary.
Done.
- Figure 1: The explanation provided in the figure legend should be condensed and must correspond precisely to the content depicted in the illustration. For example, there is no mention of the VEGF receptor or how it activates the enzyme responsible for producing LacCer, nor is it explained that the enzyme catalyzes the transfer of galactose to glucosylceramide (GlcCer) to form lactosylceramide (LacCer). Furthermore, the activation of pathways such as MAPKs, PI3K/mTOR, among others, is not defined. Additionally, the manuscript does not clarify the meaning of the abbreviations used or specify whether they refer to kinases, transcription factors, or other entities.
The figure and the caption have been edited as per the suggestion, with an explanation on the topics mentioned along with the abbreviations.
- Lines 69-71: The wording of this statement should be revised, as it is not entirely accurate; GalT-V is not a direct synonym for LacCer synthase, although it may exhibit redundant activity.
Thanks, we have remedied this issue. However, we disagree that LacCer synthase is a redundant activity of GalT-V. This function of GalT-V is not “superfluous” as this has been documented by several other groups and ours using CHO mutant mice expressing exclusively GalT-V (gift from Dr. Pam Stanley).
- Lines 111-113: It is necessary to describe this statement in greater detail, as it lacks an explicit explanation in the figure. In the illustration, the enzyme is depicted in the membrane, but it is not specified whether this refers to the plasma membrane or the Golgi apparatus. Furthermore, the manuscript does not clarify how many transferases exist (five, six, etc.) or whether all of them produce LacCer.
As described in the text (line#60) we have already described that β-1,4GalT-V and VI both can produce LacCer. We have clarified the point about the localization of β -1,4-GalT-V with the Golgi membrane in normal cells. However, we have also observed significant association of this enzyme in the cytoplasm and cell membrane in patients with colorectal cancer (Ref“Lactosylceramide synthase β-1,4-GalT-V: A novel target for the diagnosis and therapy of human colorectal cancer. Biochem. Biophys. Res. Commun. 2019, 508, 380–386”)
- Line 135: It is necessary to specify which of these NOX isoforms is being referred to: NOX1, NOX3, NOX4, NOX5, DUOX1, or DUOX2.
Our previous study showed that p47phox and p92phox protein expression was increased by LacCer (doi 10.1007/s10719-006-7920-8). Thus, p47 phox refers to NOX2(phagocytic NADPH oxidase) and p92phox refers to protein p22phox essential for the activation of several NOX complexes e.g. NOX 1,2,3 and 4.
- Lines 237-246: This paragraph lacks citations and presents ideas that are repetitive of those in the preceding paragraphs (lines 188–196).
Has been added
- Lines 247-254: This paragraph lacks citations, particularly in the first two lines.
Has been added
- Figure 4: The numbering is not consecutive, as the last figure was Figure 1. Such basic errors significantly undermine the quality of the manuscript, as they indicate that the authors did not take sufficient care to review their work prior to submission and appear to expect the reviewers to perform tasks that should have been completed beforehand. Furthermore, the manuscript presents a large number of factors, proteins, and other elements for which only abbreviations are provided, without indicating their full names or definitions.
We have fixed the figure numbering.
- Line 272: I suggest rewriting this statement, as it cites only a single study conducted in one cell line yet generalizes the findings by referring to cancer cells in the plural. Alternatively, additional references should be included to support this claim.
These studies were performed in one cell line and additional work is required.
- Line 274: The period following the word “signaling” should be removed.
Agreed, we have fixed this mistake.
- Lines 276-298: I suggest that all of these paragraphs be moved to the beginning, as they provide the context for the function of SP1. The preceding section is a specific example and, therefore, should be presented afterward.
Done
- Line 325: As Latin is a dead language, de novo should be written in italics.
Done
- Figure 3: The figure numbering should be reviewed and corrected as appropriate. Additionally, the meaning of the abbreviations is not provided. The manuscript also fails to explain what induces endothelial activation. Moreover, SP1 is not indicated in the figure in relation to the description provided.
Figure 3. Inflammatory activation of vascular endothelium during atherosclerosis. Ox-LDL and LPS (Lipopolysaccharides) promote immune cells to release cytokines like TNF-α. These cytokines activate endothelial cells and promote immune cells to the inflammation site. Sp1, although not labeled, controls expression of adhesion molecules (VCAM-1, ICAM-1), inflammatory mediators, and SR (scavenger receptors) involved in ox-LDL uptake. Figure 2 shows the downstream transcriptional pathway Sp1 moves through within the vascular endothelium. Specifically, promoting plaque formation, SRs on the surface of M1-type macrophages recognize and engulf ox-LDL to from foam cells. CD36 receptors help macrophages take in ox-LDL, which can lead to the formation of foam cells, contributing to atherosclerotic plaques in blood vessels. CD36 can also help fatty acid oxidation in anti-inflammatory M2-type macrophages, supporting tissue repair and lipid output. Simultaneously, α defensins and azurophilic granules released by neutrophils further oxidize LDL to produce more ox-LDL. Ox-LDL overproduction causes lipid accumulation and thus forms more foam cells. EC: endothelial cell; VSMC: vascular smooth muscle cell; M1: M1-type macrophages; M2: M2-type macrophages; LDL: low-density lipoprotein; ox-LDL: oxidized low-density lipoprotein; MPO: myeloperoxidase; ROS: reactive oxygen species; PMNs: polymorphonuclear neutrophils; NETs: neutrophil extracellular traps. Adapted with permission from Ref [52].
- Lines 364-366: It is important to include references in this paragraph.
For the sentence, “Sp1 modulates gene expression in endothelial cells exposed to ox-LDL, contributing to vascular injury and plaque progression,” I have added the citation: https://doi.org/10.3389/fcell.2024.1453901
- Line 421: This paragraph should be placed at the end, as it addresses the potential therapeutic applications.
We have moved the paragraph.
- Line 456: It would be highly appropriate to specify what each abbreviation refers to, as a matter of formality.
TNM (Tumor, Node, Metastasis) is now specified.
- Line 461: Figure 2 is not included in the manuscript.
Figure 2 is now added.
- Line 474: It would be important to indicate it by its abbreviation, as in human monocyte chemoattractant protein 1 (MCP-1).
Abbreviation added to text, and to the abbreviations section. - Figure 4: A brief description of the figure should be included.
Done.
- Lines 494-495: It is important to indicate the meaning of the abbreviations.
The abbreviations: cyclooxygenases-2 (COX-2), interleukin-10 (IL-10), T cells (Tregs), Transforming growth factor beta 1 (TGF-β1) have been added to the text and to the abbreviations section.
- Line 498: To what does this indication refer? Is it part of the figure legend? It is unclear.
Now changed to: “Given Sp1’s tumorigenic role, suppression of Sp1 has been shown to promote apoptosis in cancer cells [71].”
- Line 515: As in situ is a Latin term, it should be italicized.
We have removed this term.
- Line 519: It is not necessary to reference Figure 4 twice if the subsequent text also pertains to Figure 4.
Agreed, we have removed one of the figure 4 reference.
- Line 522: It is necessary to specify what E1AF refers to.
This has been rectified.
- Line 552: If it is still unknown whether Notch1 participates independently of VEGF, then what is the rationale for providing several examples of VEGF’s involvement with GalT-V? This is unclear.
This has been rectified.
- Lines 568-570: This paragraph should be combined with the preceding one, as they express the same idea; grammatically, there is no justification for separating them.
Correct, this has been done.
- Lines 572-584: There are no citations throughout this entire paragraph; may I ask why?
The information is mainly reliant on citation 73. This has been rectified.
- Lines 601-603: Is citation 2 in the first line of this paragraph the same one referenced elsewhere? Because there is no citation at the end of the paragraph, despite the statement being of significant importance.
- Line 618: It is important to specify the meaning of the abbreviation Treg, as was done for TGF-beta.
Agreed, we have added the meaning of the abbreviation.
- Line 626: The citation is missing in this paragraph; is it the same as citation 84 referenced at the beginning of the paragraph? If so, this should be clearly indicated. Failing to specify citations reflects a significant lack of attention to detail.
Yes, it is citation 80.
- Figure 5: Please review the figure numbering, as the consecutive sequence is incorrect. Several proteins and enzymes depicted in the figure, such as calcineurin, c-Jun, Disheveled, AP-1, among others, are not described in the figure legend.
Done
- Line 649: I suggest removing this subtitle, as it is unnecessary; it is understood that a background of the pathway will be provided.
Agreed, we have removed that subtitle
- Lines 650-654: The paragraph lacks citations.
Agreed, we appreciate the reviewer’s comment. We have added the appropriate citations
- Line 660: The figure numbering does NOT correspond correctly to the sequence
Agreed, we have fixed that mistake.
- Lines 700-706: Citations are missing! It is highly concerning that references are not provided throughout the manuscript in a review article, as this reflects significant negligence and lack of diligence.
Agreed, we appreciate the reviewer’s comment. We have added the appropriate citations.
- Lines 735-736: The statement lacks citations!
Agreed, we appreciate the reviewer’s comment. We have added the appropriate citation.
- Line 752: I suggest removing this subtitle, as it is unnecessary; it is understood that a background of the pathway will be provided.
Agreed!
- Line 769: Please review the figure numbering.
Agreed, we have fixed the figure numbering
- Lines 785-788: Please review the figure numbering. Additionally, this paragraph conveys the same idea as the preceding one and, therefore, should not be separated but included together with the previous paragraph.
Agreed, we have combined both paragraphs.
- Lines 794-799: The same recommendation as above applies in all respects.
Done
- Lines 818-824: References are missing in this paragraph; these are important assertions.
Agreed, we have fixed added the references.
- Lines 840-841: This paragraph is isolated and should be combined with the following one, as they convey the same contextual idea.
The paragraph has been edited, and has been combined with the intro of Wnt
- Lines 842-843: It is important to specify the duration for which the drug was administered in this study.
We have added this detail.
- Figure 6: Please review the figure numbering and ensure it is consistent with that of the other figures. Several abbreviations present in the figure remain undefined, and a concise and appropriate description of the figure is lacking.
We have the figure numbering. We have also elaborated on the figure description and defined the abbreviations.
- Line 769: Please review the figure numbering.
We have fixed the figure numbering
- Lines 969-971: It would be advisable for the authors to provide a more detailed description of the study, as the current explanation is quite vague.
A detailed description has been provided.
- Lines 994-995: This paragraph is part of the same idea as the following paragraph.
The paragraph has been edited
- Line 1036: There is a double period; please review.
Agreed, we have fixed the mistake.
- Figure 7: Please review the figure numbering and provide a more accurate description of the figure in accordance with its content.
Done
- Line 1074: A citation is missing for the statement in the first line.
Agreed, we appreciate the reviewer’s comment. We have added the appropriate citation.
- Figure 9: Please review the figure numbering.
We have fixed the figure numbering.
- Line 1086: Why was this topic not addressed when discussing the Wnt pathway? It seems out of context and repetitive, as several points mentioned here were already covered earlier during the discussion of the Wnt pathway.
Agreed, Like PD-L1 protein Wnt is also a glycoprotein, and we would like to present the PD-L1 section separately by explaining its protein structure at molecular level and role of glycosylation of PD-L1 protein with reference to its function and stability.
- Lines 1210-1216: No references are provided in this paragraph. May I ask why?
We have added the references in this paragraph.
- Lines 1225-1229: The assertion regarding the relationship between P62, autophagy, and the persistent inflammatory state in chronic smokers is unclear.
Studies show that in mice exposed to cigarette smoke-induced- emphysema and lung tissue from COPD patients have markedly increased levels of LacCer with increasing severity of emphysema. This was accompanied with an increase in the protein expression of p62, a biomarker for defective autophagy. Conversely, treatment with D-PDMP; a GlcCer/LacCer synthase inhibitor ,controlled disease pathology and defective autophagy in mice exposed to cigarette smoke, and in bronchial epithelial cells
- Line 1247: Why are only prostate and breast cancers mentioned? Would these findings not be applicable to other types of cancer? The authors place significant emphasis on colorectal cancer.
Prostate and breast cancers were mentioned as they contribute heavily to mortality. Indeed, there are many other cancers wherein β-1,4-GalT-V is overexpressed.
- Line 1249: In the conclusion, the authors do not mention other pathways analyzed, such as autophagy, Snail1, Hedgehog, LDL, and atherosclerosis. The authors should strive to condense all the information, as they address a large number of topics, yet not all are included in the conclusion.
Agreed, this issue has been addressed, thanks!
- Line 1284: Table 1 is not referenced in the text.
We have referenced the table in the text.
Reviewer 2 Report
Comments and Suggestions for Authors
Reviewer Comments:
- Abstract is comprehensive but overly dense. Consider breaking long sentences into clearer statements and reducing repetition. Emphasizing distinct sections—background, findings, and implications—would enhance clarity for broader readership.
- The manuscript covers a vast range of interconnected topics. While each section is detailed, the transitions between Sp1, GalT-V, Notch-1, Wnt signaling, and their downstream effects are occasionally abrupt. A summarizing table or flowchart delineating these interactions could improve reader navigation and comprehension.
- In the Sp1 discussion is thorough, but several points—particularly those concerning O-GlcNAcylation and its transcriptional implications—are repeated in different sections. Consolidating these points could enhance readability and avoid redundancy.
- While the manuscript reviews numerous interactions (e.g., GalT-V–Notch-1, GalT-V–Wnt-1), the mechanistic evidence supporting some interactions (e.g., direct galactosylation of Wnt components) is not well substantiated with experimental data. Suggest clarifying whether these are hypothesized or experimentally validated.
- Concept of a VEGF-independent pathway involving GalT-V/LacCer and Notch-1 is intriguing and well-presented. However, further clarification of whether this pathway has been validated in vivo across different cancer models is necessary to assess its translational significance.
- Present review appropriately suggests GalT-V as a potential drug target. However, the therapeutic specificity and potential off-target effects of GalT-V inhibition (e.g., with D-PDMP) warrant deeper exploration. Discussion on pharmacological challenges would strengthen this section.
- In GalT-V section, the claim that GalT-V can serve as a diagnostic biomarker for colorectal cancer is compelling but requires further elaboration. Is GalT-V expression tissue-specific or detectable in biofluids (e.g., blood, urine)? Data from patient cohorts would enhance this claim.
- While the manuscript includes well-designed figures, there are instances where the figure legends could more clearly correlate with the main text. Ensure that all figures (e.g., Figure 3 and Figure 6) are consistently referenced and discussed at appropriate places in the manuscript.
- The role of diet, smoking, and obesity on GalT-V expression is acknowledged but underdeveloped. Incorporating specific studies or data linking environmental exposures to GalT-V regulation would support this point more robustly.
- Connection between lipid metabolism (via FASN) and Sp1 transcriptional activity is novel but underexplored. Suggest expanding this section with additional context or evidence, particularly how it might integrate with GalT-V signaling.
- Manuscript notes the interaction between Sp1 and PD-L1 transcription. It would be beneficial to develop further how GalT-V-driven glycosylation or metabolic shifts may indirectly modulate immune checkpoint expression or response to ICIs.
- There are inconsistent uses of terminology (e.g., GalT-V vs β-1,4-GalT-V; NICD vs NECD without redefinition). A glossary or consistency check would improve clarity for readers unfamiliar with the field.
- The review largely supports the GalT-V axis without acknowledging potential contradictory findings or limitations in current knowledge. A brief paragraph on known controversies or unanswered questions would provide a more balanced and critical perspective.
- While the manuscript demonstrates a high level of scientific understanding, the use of English requires refinement for better clarity and flow. Several sentences are overly long, contain redundant phrases, or use technical terms without proper syntactic alignment. A thorough proofreading is recommended to improve grammar, punctuation, and readability throughout the manuscript.
15. In the conclusion section would benefit from a more focused summary of future directions. For example, prioritizing which pathways (Sp1, Notch, Wnt) represent the most promising for targeted drug development, or highlighting whether biomarker-based stratification is a feasible next step for clinical trials.
While the manuscript provides an extensive and insightful overview of the GalT-V interactome and its involvement in key pathological processes, some sections would benefit from improved clarity, more concise presentation, and grammatical refinement. Specific comments regarding structural flow, technical depth, figure integration, and language usage have been provided in the detailed review. Addressing these minor issues will significantly enhance the manuscript's readability and impact without requiring substantial changes to the core content.
Based on the scientific merit, depth of discussion, and novelty of the content presented in the manuscript, I recommend a minor revision of the present manuscript.
Comments on the Quality of English Language
While the manuscript demonstrates a high level of scientific understanding, the use of English requires refinement for better clarity and flow. Several sentences are overly long, contain redundant phrases, or use technical terms without proper syntactic alignment. A thorough proofreading is recommended to improve grammar, punctuation, and readability throughout the manuscript
Author Response
- Abstract is comprehensive but overly dense. Consider breaking long sentences into clearer statements and reducing repetition. Emphasizing distinct sections—background, findings, and implications—would enhance clarity for broader readership.
Agreed! We have edited the abstract.
- The manuscript covers a vast range of interconnected topics. While each section is detailed, the transitions between Sp1, GalT-V, Notch-1, Wnt signaling, and their downstream effects are occasionally abrupt. A summarizing table or flowchart delineating these interactions could improve reader navigation and comprehension.
Thanks, this is a great suggestion. However, this maybe beyond the scope of this review article.
- In the Sp1 discussion is thorough, but several points—particularly those concerning O-GlcNAcylation and its transcriptional implications—are repeated in different sections. Consolidating these points could enhance readability and avoid redundancy.
Agreed, we have edited 5.1.3.
- While the manuscript reviews numerous interactions (e.g., GalT-V–Notch-1, GalT-V–Wnt-1), the mechanistic evidence supporting some interactions (e.g., direct galactosylation of Wnt components) is not well substantiated with experimental data. Suggest clarifying whether these are hypothesized or experimentally validated.
For the GalT-V and Wnt-1 interaction, we added that the interaction is speculated.
- Concept of a VEGF-independent pathway involving GalT-V/LacCer and Notch-1 is intriguing and well-presented. However, further clarification of whether this pathway has been validated in vivo across different cancer models is necessary to assess its translational significance.
This tenet has not been validated across several types of cancer. Clearly, additional studies are justified to propose a broad clinical significance
- Present review appropriately suggests GalT-V as a potential drug target. However, the therapeutic specificity and potential off-target effects of GalT-V inhibition (e.g., with D-PDMP) warrant deeper exploration. Discussion on pharmacological challenges would strengthen this section.
Agreed!
- In GalT-V section, the claim that GalT-V can serve as a diagnostic biomarker for colorectal cancer is compelling but requires further elaboration. Is GalT-V expression tissue-specific or detectable in biofluids (e.g., blood, urine)? Data from patient cohorts would enhance this claim.
This is a great suggestion! Our follow up studies shall elaborate on this point in the near future
- While the manuscript includes well-designed figures, there are instances where the figure legends could more clearly correlate with the main text. Ensure that all figures (e.g., Figure 3 and Figure 6) are consistently referenced and discussed at appropriate places in the manuscript.
We have fixed the figure numbering and have referenced them in the text.
- The role of diet, smoking, and obesity on GalT-V expression is acknowledged but underdeveloped. Incorporating specific studies or data linking environmental exposures to GalT-V regulation would support this point more robustly.
Agreed! Our goal here is to present whatever information we have currently to bring awareness to this exciting area of research. When additional information on these lines of investigation appears, we will be happy to discuss in a follow-up article.
- Connection between lipid metabolism (via FASN) and Sp1 transcriptional activity is novel but underexplored. Suggest expanding this section with additional context or evidence, particularly how it might integrate with GalT-V signaling.
Again, this is a great suggestion! However, due to space limitations, we can present this material elsewhere.
- Manuscript notes the interaction between Sp1 and PD-L1 transcription. It would be beneficial to develop further how GalT-V-driven glycosylation or metabolic shifts may indirectly modulate immune checkpoint expression or response to ICIs.
Thanks for this suggestion.
- There are inconsistent uses of terminology (e.g., GalT-V vs β-1,4-GalT-V; NICD vs NECD without redefinition). A glossary or consistency check would improve clarity for readers unfamiliar with the field.
We have fixed these mistakes.
- The review largely supports the GalT-V axis without acknowledging potential contradictory findings or limitations in current knowledge. A brief paragraph on known controversies or unanswered questions would provide a more balanced and critical perspective.
Thanks for this suggestion.
- While the manuscript demonstrates a high level of scientific understanding, the use of English requires refinement for better clarity and flow. Several sentences are overly long, contain redundant phrases, or use technical terms without proper syntactic alignment. A thorough proofreading is recommended to improve grammar, punctuation, and readability throughout the manuscript.
We have attempted to rectify this issue in the edited version.
- In the conclusion section would benefit from a more focused summary of future directions. For example, prioritizing which pathways (Sp1, Notch, Wnt) represent the most promising for targeted drug development, or highlighting whether biomarker-based stratification is a feasible next step for clinical trials.
Indeed, we have edited the conclusion in view of this suggestion.
Reviewer 3 Report
Comments and Suggestions for Authors
This review article, "The GalT-V Interactome: A Network of Pathways Driving Cardiovascular Cancer and Inflammatory Diseases", aims to systematically summarize the mechanistic roles of β-1,4-galactosyltransferase V (GalT-V) in various pathological conditions such as cancer, cardiovascular diseases, and inflammatory disorders. The manuscript focuses on GalT-V–mediated lactosylceramide (LacCer) production and its regulatory interactions with key signaling pathways including Sp1, Notch-1, and Wnt. The topic is timely and innovative, particularly in its attempt to connect GalT-V, a glycosyltransferase enzyme, to tumor immunology, cardiovascular injury, metabolic reprogramming, and cancer stem cell regulation. This integrative perspective is of great interest to the field.
However, the manuscript suffers from significant structural redundancy, verbose language, inconsistent logic, and non-uniform terminology. Substantial revisions are necessary to meet the academic standards of the *International Journal of Molecular Sciences*.
Major Comments:
-
The abstract is overly long and should be significantly condensed** to highlight only the most essential information and conclusions.
-
Some mechanistic descriptions are vague, particularly in Section 5.8, where the association between PD-L1 glycosylation and GalT-V is speculative and lacks direct experimental support. Please clearly distinguish validated mechanisms from hypothetical ones, using cautious language such as “may” or “possibly.”
-
The proposed **indirect regulation of Sp1 O-GlcNAcylation by GalT-V (Section 5.1) is not convincingly explained. The logical link is incomplete and requires supporting evidence—such as metabolomics data—to demonstrate how GalT-V affects the UDP-GlcNAc pool or hexosamine biosynthetic pathway.
-
Sections 5.4 (GalT-V and Wnt-1) and 5.9 (β-catenin) are repetitive and scattered. I recommend merging them into a unified subsection titled “Wnt/β-catenin signaling”, with a focused discussion on how GalT-V-mediated glycosylation modulates this pathway.
-
The therapeutic strategies listed in Table 1 are not well integrated into the main text. For instance, Ipafricept, which targets FZD8, is listed in the table but not discussed in Section 5.5. Please ensure all entries are substantiated in the relevant sections of the manuscript.
-
The conclusion section summarizes key findings but does not adequately reflect the field's current challenges—such as tissue-specific toxicity or resistance mechanisms. Please add a forward-looking discussion of future directions.
-
Figure 9 lacks clarity regarding the function of the ICD. It mentions RNA binding but fails to connect this to immune regulation or other functional pathways.
-
In molecular mechanism figures, several key regulatory arrows are missing. For example, the transcriptional activation of GalT-V by Sp1 is not annotated in the diagrams.
-
Abbreviations should be standardized and consistent throughout the manuscript.
-
All abbreviations (e.g., PCP, CaMKII) should be defined upon first use. For example, in Section 4.2, “Wnt/PCP pathway” is introduced without full explanation.
-
“Snail-1” and “Snail1” are used inconsistently in Section 5.7. Please use a consistent format.
-
The figures are complex and data-rich, but the accompanying textual explanations are insufficient. The manuscript should provide more guidance to help readers interpret each panel.
-
Some key conclusions lack citations from recent literature, especially studies published within the last 2–3 years.
-
Reference formatting is inconsistent and needs to be carefully revised to follow journal guidelines.
-
The manuscript repeats the roles of Sp1, Notch-1, and Wnt across multiple sections, resulting in a fragmented and disorganized structure. A clearer thematic organization is needed. It is recommended to emphasize GalT-V as a central regulatory node linking metabolism, epigenetics, and immunity, and to propose multi-target therapeutic strategies that address this interactome holistically.
Author Response
- The abstract is overly long and should be significantly condensed** to highlight only the most essential information and conclusions.
Agreed! We have considerably shortened the abstract.
- Some mechanistic descriptions are vague, particularly in Section 5.8, where the association between PD-L1 glycosylation and GalT-V is speculative and lacks direct experimental support. Please clearly distinguish validated mechanisms from hypothetical ones, using cautious language such as “may” or “possibly.”
Agreed, we fixed the format.
- The proposed **indirect regulation of Sp1 O-GlcNAcylation by GalT-V (Section 5.1) is not convincingly explained. The logical link is incomplete and requires supporting evidence—such as metabolomics data—to demonstrate how GalT-V affects the UDP-GlcNAc pool or hexosamine biosynthetic pathway.
Agreed, we have edited this description in view of your suggestion. We added that supporting evidence, such as metabolomics data is necessary to demonstrate whether β-1,4-GalT-V inhibition affects UDP-GlcNAc levels and nuclear glycosylation.
- Sections 5.4 (GalT-V and Wnt-1) and 5.9 (β-catenin) are repetitive and scattered. I recommend merging them into a unified subsection titled “Wnt/β-catenin signaling”, with a focused discussion on how GalT-V-mediated glycosylation modulates this pathway.
We do respect the reviewer’s comment. However, we feel that the b-catenin should stand as its own section.
- The therapeutic strategies listed in Table 1 are not well integrated into the main text. For instance, Ipafricept, which targets FZD8, is listed in the table but not discussed in Section 5.5. Please ensure all entries are substantiated in the relevant sections of the manuscript.
Ipafricept has been discussed in Section5.4.3.
- The conclusion section summarizes key findings but does not adequately reflect the field's current challenges—such as tissue-specific toxicity or resistance mechanisms. Please add a forward-looking discussion of future directions.
Thanks! We have edited this section.
- Figure 9 lacks clarity regarding the function of the ICD. It mentions RNA binding but fails to connect this to immune regulation or other functional pathways.
Agreed, and explained
- In molecular mechanism figures, several key regulatory arrows are missing. For example, the transcriptional activation of GalT-V by Sp1 is not annotated in the diagrams.
The transcriptional activation of Sp1 by GalT-V has been indicated at the bottom of the figure. Also, all the arrows have been pointed in the figure.
- Abbreviations should be standardized and consistent throughout the manuscript.
Agreed, we have fixed the abbreviations.
- All abbreviations (e.g., PCP, CaMKII) should be defined upon first use. For example, in Section 4.2, “Wnt/PCP pathway” is introduced without full explanation.
Agreed, we have fixed these mistakes and made sure all the abbreviation are defined at first use.
- “Snail-1” and “Snail1” are used inconsistently in Section 5.7. Please use a consistent format.
Agreed, we fixed the format.
- The figures are complex and data-rich, but the accompanying textual explanations are insufficient. The manuscript should provide more guidance to help readers interpret each panel.
Thanks, in view of this suggestion, we have edited various Fig’s
- Some key conclusions lack citations from recent literature, especially studies published within the last 2–3 years.
We greatly appreciate your forwarding recent reports so we may include them.
- Reference formatting is inconsistent and needs to be carefully revised to follow journal guidelines.
We have fixed the reference formatting to follow the journal guidelines.
- The manuscript repeats the roles of Sp1, Notch-1, and Wnt across multiple sections, resulting in a fragmented and disorganized structure. A clearer thematic organization is needed. It is recommended to emphasize GalT-V as a central regulatory node linking metabolism, epigenetics, and immunity, and to propose multi-target therapeutic strategies that address this interactome holistically.
We greatly appreciate this valuable suggestion. In Fig1 we have attempted to achieve the proposal made by this reviewer. However, as more information becomes available, it will help us address these items in greater detail.
Round 2
Reviewer 1 Report
Comments and Suggestions for Authors
The authors have adequately addressed all of my comments and made the necessary adjustments to the figures, figure legends, and the conclusion. Furthermore, they revised the title, which now more accurately reflects the content and scope of the review
Reviewer 3 Report
Comments and Suggestions for Authors
The authors have solved the problem well. I think this version of the manuscript can be published in IJMS.